# Can an InChI for Nano Address the Need for a Simplified Representation of Complex Nanomaterials across Experimental and Nanoinformatics Studies?

**DOI:** 10.3390/nano10122493

**Published:** 2020-12-11

**Authors:** Iseult Lynch, Antreas Afantitis, Thomas Exner, Martin Himly, Vladimir Lobaskin, Philip Doganis, Dieter Maier, Natasha Sanabria, Anastasios G. Papadiamantis, Anna Rybinska-Fryca, Maciej Gromelski, Tomasz Puzyn, Egon Willighagen, Blair D. Johnston, Mary Gulumian, Marianne Matzke, Amaia Green Etxabe, Nathan Bossa, Angela Serra, Irene Liampa, Stacey Harper, Kaido Tämm, Alexander CØ Jensen, Pekka Kohonen, Luke Slater, Andreas Tsoumanis, Dario Greco, David A. Winkler, Haralambos Sarimveis, Georgia Melagraki

**Affiliations:** 1School of Geography, Earth and Environmental Sciences, University of Birmingham, Edgbaston, Birmingham B15 2TT, UK; a.papadiamantis@bham.ac.uk; 2Nanoinformatics Department, NovaMechanics Ltd., 1666 Nicosia, Cyprus; afantitis@novamechanics.com (A.A.); melagraki@nvemechanics.com (A.T.); 3Edelweiss Connect GmbH, Hochbergerstrasse 60C, 4057 Basel, Switzerland; thomas.exner@edelweissconnect.com; 4Department Biosciences, Paris Lodron University of Salzburg, Hellbrunnerstrasse 34, 5020 Salzburg, Austria; Martin.Himly@sbg.ac.at; 5School of Physics, University College Dublin, Belfield, Dublin 4, Ireland; vladimir.lobaskin@ucd.ie; 6School of Chemical Engineering, National Technical University of Athens, 157 80 Athens, Greece; filipposd@gmail.com (P.D.); irini.liampa@gmail.com (I.L.); hsarimv@central.ntua.gr (H.S.); 7Biomax Informatics AG, Robert-Koch-Str. 2, 82152 Planegg, Germany; dieter.maier@biomax.com; 8National Health Laboratory Services, 1 Modderfontein Rd, Sandringham, Johannesburg 2192, South Africa; natashas@nioh.ac.za (N.S.); maryg@nioh.ac.za (M.G.); 9QSAR Lab Ltd., Aleja Grunwaldzka 190/102, 80-266 Gdansk, Poland; a.rybinska@qsarlab.com (A.R.-F.); m.gromelski@qsarlab.com (M.G.); t.puzyn@qsarlab.com (T.P.); 10Department of Bioinformatics—BiGCaT, School of Nutrition and Translational Research in Metabolism, Maastricht University, Universiteitssingel 50, 6229 ER Maastricht, The Netherlands; egon.willighagen@maastrichtuniversity.nl; 11Department Chemicals and Product Safety, Federal Institute for Risk Assessment, Max-Dohrn-Str. 8-10, 10589 Berlin, Germany; Blair.Johnston@bfr.bund.de; 12Haematology and Molecular Medicine, University of the Witwatersrand, 1 Jan Smuts Ave, Johannesburg 2000, South Africa; 13UK Centre for Ecology and Hydrology, Maclean Building, Benson Lane, Crowmarsh Gifford OX10 8BB, UK; martzk@ceh.ac.uk (M.M.); amagre@ceh.ac.uk (A.G.E.); 14LEITAT Technological Center, Circular Economy Business Unit, C/de La Innovació 2, 08225 Terrassa, Barcelona, Spain; nbossa@leitat.org; 15Faculty of Medicine and Health Technology, Tampere University, FI-33014 Tampere, Finland; angela.serra@tuni.fi (A.S.); dario.greco@tuni.fi (D.G.); 16School of Chemical, Biological, and Environmental Engineering, Oregon State University, 116 Johnson Hall 105 SW 26th St., Corvallis, OR 97331, USA; Stacey.Harper@oregonstate.edu; 17Institute of Chemistry, University of Tartu, Ülikooli 18, 50090 Tartu, Estonia; karu@ut.ee; 18The National Research Center for the Work Environment, Lersø Parkallé 105, 2100 Copenhagen, Denmark; alj@nfa.dk; 19Misvik Biology OY, Karjakatu 35 B, 20520 Turku, Finland; pkpekka@gmail.com; 20Institute of Cancer and Genomics, University of Birmingham, Edgbaston, Birmingham B15 2TT, UK; luke.slater.1@bham.ac.uk; 21Institute of Molecular Sciences, La Trobe University, Kingsbury Drive, Bundoora 3086, Australia; D.Winkler@latrobe.edu.au; 22Monash Institute of Pharmaceutical Sciences, Monash University, Parkville 3052, Australia; 23School of Pharmacy, University of Nottingham, Nottingham NG7 2RD, UK; 24CSIRO Data61, Pullenvale 4069, Australia

**Keywords:** molecular structure, machine-readable, nanomaterials descriptors, core, surface, surface functionalization, complex nanostructures

## Abstract

Chemoinformatics has developed efficient ways of representing chemical structures for small molecules as simple text strings, simplified molecular-input line-entry system (SMILES) and the IUPAC International Chemical Identifier (InChI), which are machine-readable. In particular, InChIs have been extended to encode formalized representations of mixtures and reactions, and work is ongoing to represent polymers and other macromolecules in this way. The next frontier is encoding the multi-component structures of nanomaterials (NMs) in a machine-readable format to enable linking of datasets for nanoinformatics and regulatory applications. A workshop organized by the H2020 research infrastructure NanoCommons and the nanoinformatics project NanoSolveIT analyzed issues involved in developing an InChI for NMs (*NInChI*). The layers needed to capture NM structures include but are not limited to: core composition (possibly multi-layered); surface topography; surface coatings or functionalization; doping with other chemicals; and representation of impurities. NM distributions (size, shape, composition, surface properties, etc.), types of chemical linkages connecting surface functionalization and coating molecules to the core, and various crystallographic forms exhibited by NMs also need to be considered. Six case studies were conducted to elucidate requirements for unambiguous description of NMs. The suggested *NInChI* layers are intended to stimulate further analysis that will lead to the first version of a “nano” extension to the InChI standard.

## 1. Introduction

Nanotechnology is a key enabling technology to improve the durability and function of diverse products in the skincare, electronics, textiles, sporting equipment, medical diagnostics, and therapeutics industries. The Project on Emerging Technologies (PEN) has identified >1800 nanotechnology-based products [1]. The rapid rise in adoption of nanotechnology by industry has fueled concerns over the safety of nanomaterials. This has driven a concomitant rise in nanosafety research and development of safe-by-design (SbD) principles that have been a major theme of several framework programs of the European Commission for the past decade [2,3]. During this time, a transition from pure in vivo toxicology studies to more mechanistic in vitro studies occurred that involved advanced organotypic cell culture models mimicking the relevant human body barriers, i.e., inhalation and ingestion. The transition to mechanistic approaches also coincided with a rise in computational models, largely machine learning-based, that augment, leverage and replace in vitro and scarce in vivo data for regulatory purposes. These changes were driven by the “3R principle”—reduction, refinement, and replacement of animal experiments [4,5]. Development of high-throughput screening to greatly increase the amount of data available for regulators and for training hazard prediction tools was advocated during Horizon 2020 [6,7]. When combined with development of physicochemical characterization methods, exposure models, and life cycle assessment, this evolution provides a solid foundation for SbD strategies and nanomaterials (NM) risk management that supports sustainable innovation [8,9]. However, a commonly cited roadblock is lack of curated, organized and harmonized datasets for NMs toxicity [10], like those available for small molecules on UniChem, ChEMBL and other platforms. 

Central to the success of small molecule databases and chemoinformatics approaches has been the availability of unique structural representations for each chemical. These distinguish one chemical compound from another and facilitate unambiguous mapping of chemical substances from different sources, enabling automated data retrieval and exploitation. Beyond the chemical formula nomenclature systems, such as IUPAC’s naming approach for organic chemicals, aim to provide human-readable information on the spatial organization of the atoms, by having an agreed order for identifying branches and functional groups. However, these become very complex for even moderate size chemicals and are complicated to parse for humans and computers, so trivial names are often used that can lead to ambiguities. CAS registry numbers (RNs), for example, are unique numerical identifiers assigned by the Chemical Abstracts Service (CAS) to every chemical substance described in the open scientific literature but, like trivial names, are not systematically connected to structure. Clearly, the cheminformatics field needed a machine-readable representation of chemical structural information. The common line notation formats for describing chemical structures, such as the simplified molecular-input line-entry system (SMILES) [11] and the IUPAC International Chemical Identifier (InChI) [12,13], were devised to describe the chemistry and connectivity of a molecule. Here, we refer to these and the InChI for NMs introduced here as structural representations (or linear notation) to separate them from chemical identifiers like CAS RN where information on the substance requires querying the corresponding registries. Note that InChI and SMILES were developed for chemical compounds that can be accurately and completely described by a chemical graph, whose vertices correspond to atoms and edges correspond to chemical bonds in the compound. This line notation can be used to build the 2 D chemical structure only, as the 3 D structure depends the relative energies of conformations in specific solvents. 

Clearly, NMs cannot be represented as a chemical graph and pose a conceptual challenge to development of a machine-readable structural representation. Their 3 D structures, the close correlation between size and properties, the emergence of quantum properties at sub-30 nm scales [14], and their very high surface to volume ratios relative to the bulk, means that NM representations requires information on their chemical composition, physical, and structural features. An additional complication is the dynamic nature of their properties that are influenced by environment, changing with composition (salts, biomolecules), temperature, time, or other external factors. Thus, NMs have both *intrinsic* properties (inherent to the material) and *extrinsic* properties that depend on environmental conditions [15]. Thus, a text representation for NMs must include information on its size, shape, internal structure, and surface characteristics that may include molecules chemically attached to the surface. As the toxicity of NMs is largely due to their surface properties, it is essential that any representation encodes such information. Other materials also present challenges for encoding their structure—macromolecules, polymers, porous and 2 D materials, and mixtures. The research community has been developing extensions to the InChI for encoding these more complex materials that provide important insights into how to represent NMs. 

### 1.1. Progress in InChI Representation of Small Molecules, Polymers, Mixtures and Reactions

The increasing capabilities of in silico tools to model and analyze chemical information in diverse molecular and materials sciences, including toxicology and risk assessment, provide a strong incentive to develop machine-readable formats for substances. Unique representations are of essential for distinguishing chemicals, comparing chemical substance datasets from different sources, and retrieval and exploitation of data. To this end, the InChI has been developed by the International Union of Pure and Applied Chemistry (IUPAC), with contributions from the U.S. National Institute of Standards and Technology (NIST) and the InChI Trust. Chemical repositories such as PubChem [16], ChEMBL [17] and ChemSpider [18] have incorporated InChI into their search engines. 

The InChI is a non-proprietary, Open Source chemical notation that encodes chemical features in a hierarchically manner. The InChI is structure-based so that anyone can produce the InChI from the structure of a chemical substance by following the specifications provided by the InChI Trust. InChIs are unique to each form of a substance and complement the CAS RNs that are the same for all forms of a substance. For example, silicon dioxide has numerous forms (colloidal, fumed, aerogel, gel etc.) depending on the synthesis route, each of which has different properties but same CAS RN. 

The InChI follows a layered approach to reflect the various features of a chemical structure. The major InChI layers are: *Main*; *Charge*; *Stereochemical*; *Isotopic*; and *Fixed-H*, each of which may have associated sublayers. In addition, the *Reconnected* layer is an optional extension for representing metal-containing compounds. In the *Main* layer, information on composition and skeletal connection between atoms and hydrogens for the core structure are represented. The *Charge* layer encodes the net charge, the *Stereochemistry* layer represents geometric and optical isomerism, the *Isotopic* layer identifies any isotopically labeled atoms, and the *Fixed-H* layer specifies tautomers. Details on construction of InChIs can be found in an introductory publication [13]. InChIs can also accommodate two or more disconnected molecules. The InChI is generated by software and is complemented by its compact (hashed, 27 character) representation, the InChIKey, to facilitate web-searching. Since NMs in application environments are often stabilized by grafted small molecules, our initial idea is to use existing InChI for the surface ligands and add it to an InChI that describes the NMs core. 

A hierarchical, layered structure is sufficiently flexible to accommodate additional features of molecules, more complex molecular systems, and the complexity of NMs. Useful precedents exist, as new InChI layers have been defined to uniquely represent chemical reactions, inorganic molecules, large macromolecules such as polymers, or even mixtures of chemicals. The way InChIs have been extended to polymers and mixtures provides useful guidance on how non-structural properties can be encoded in a NM InChI (*NInChI)*, as discussed below. 

Chemical reactions are encoded by a non-proprietary, international notation, Reactions InChI (RInChI) [19], further refined [20]. Reactions consist of molecules, synthesis conditions (times, temperatures, concentrations, rates of mixing, yields), and numerous other quantities [20]. RInChI is a unique machine-readable character string that describes chemical reactions and is suitable for data storage and indexing. The RInChI consist of 6 layers separated by/, as in the standard InChI, but also with combinations of <, > and <> symbols. The < symbol separates compounds in one group (reactants, products, catalysts) and the > symbol separate the groups. The first layer contains the version number and the second and third layers contain the reactants and products whose structures are represented by InChIs in alphabetical order and separated by the ! symbol. Layer four includes all structures present at both the beginning and end of the reaction (i.e., catalysts, solvents, etc.). A flag in layer five distinguishes reactants from products. If layer two contains the reactants a + symbol is used, if layer 3 contains the reactants a—symbol is used, with the = symbol used to indicate an equilibrium. Finally, layer six contains the number of distinct non-structural substances (e.g., enzymes, natural substances, heterogeneous metals, etc.) that are encoded in layers 2–4. An RInChI can still be valid even if no structures are included in layers 2 or 3 or in both. This indicates that these components are unknown or that non-structural substances are included whose information is encoded in layer six. Additionally, there is one simple modifier to show in which direction the reaction goes (/d+, /d− or /d=). When defining *NInChI*s, separation of different substances by ! and <> symbols may be useful to represent multi-constituent NMs, such as core-shell NMs or non-symmetrical NMs [21]. 

InChI version 1.05 includes a beta feature for representing polymers, the PInChI. This includes a new ‘polymer’ layer that encodes whether a source-based or structure-based representation is used for the monomer unit, and how the monomers are connected. There is so far no consensus on how to include specific end groups or the degree of polymerization (i.e., the number of repeating units). The InChI polymer working group decided to treat this as metadata to be used as an additional search query in conjunction with the PInChI. As many NMs are coated with polymers to provide steric stabilization, increased solubility, or reduced protein binding or toxicity etc. [22], we proposed using existing PInChIs to describe the polymer layer, with an indicator for how the polymer is attached (physically adsorbed, grafted onto the NM surface, or via S–S bonds for example). As with small molecule ligands, we simply add the different InChIs together and provide an order to show that the structure reads from the core to the outermost surface. We describe this approach further in the case studies below. 

An alternative to PInChI, BigSMILES, for transforming polymer information into machine-readable form was recently introduced by Lin et al. [23]. The standard SMILES specification formed the basis and additional symbols were defined to encode polymer peculiarities due to their intrinsically stochastic nature. In comparison with discrete small molecules, polymers do not have a single, well-defined chemical structure because they consist of a distribution of different chain lengths, blocks, and cross linkages. BigSMILES avoids these obstacles by encoding the polymeric fragments with curly brackets, the structures of the repeating units by the SMILES syntax, and bonding symbols that specify the chemical connections to form the polymers. Again, properties like degree of polymerization and nature of end groups are not explicitly specified even though they have significant influence on the properties of the polymer. CurlySMILES extends SMILES further with annotations for storage, retrieval and modeling of interlinked, coordinated, assembled and adsorbed molecules in supramolecular structures and nanodevices [24].

Finally, the recently developed notation for mixtures, MInChI, identifies three main elements as being essential—compounds, quantities, and hierarchy each of which is encoded as a layer of the MInChI [25]. Although not yet part of the standard InChI definition, an open source editor is available that includes several thousand examples. Since the MInChI describes the dispersion of one substance in another and the weight % of the solute, we can adapt it for *NInChI* in two different ways: to describe cores composed of different chemical components and; to incorporate information about solutions (wt%) for frequently encountered NM dispersions.

### 1.2. A Proposal for a Hierarchical Representation of Chemical and Structural Complexity of NMs 

NMs are broadly classified as organic, inorganic, or carbon-based. Organic NMs include lipid- or protein-based NMs [26], inorganic NMs include metal- and metal oxide-based materials with at least one dimension in the range 1–100 nm [27], and carbon-based includes graphene and fullerene materials [28] and polymeric materials broadly. Carbon-based NMs also include biodegradable and biocompatible NMs (used in nanomedicine and often larger than 100 nm) and biopersistent materials [29]. This diversity of chemical compositions, within and between NM classes, and the presence of surfaces with diverse modifications distinguished NMs from their bulk chemical constituents. Moreover, the surface chemistry of NMs largely determines their extrinsic properties, activities, and reactivities. Creation of surfaces with bespoke functionalities is therefore central to the industrial and biological applications of NMs, and also impacts on their toxicity and pathogenicity. The influence of nanoscale 2 D and 3 D surface topographies of (nano) materials on their biological properties (biomolecules and cells) has recently been emphasized [30,31]. 

NMs are thus characterized by complex structures and chemistries [32], that can undergo extensive transformations upon interaction with different media during their life cycles [33,34], include agglomeration [30] and dynamic corona formation in different environments [35,36,37]. The high surface activity of NMs has a strong impact on their hazard profiles and biological functions [38,39]. Building on the achievements of large consortia addressing safety of NMs, the Organisation for Economic Cooperation and Development (OECD) has agreed on a workflow for risk assessment based on asking three crucial questions, “What are they?” (phase 1), “Where do they go?” (phase 2), and “What do they do?” (phase 3). To be consistent with the OECD’s “Physical-chemical decision framework to inform decisions for risk assessment of manufactured nanomaterials” [40] and to rationally describe the complexity of NMs in a structured way, a hierarchical description of NMs from the inside out is required (see Figure 1).

Tier 1 includes all relevant information on a chemical substance, encompassing the composition of a NM. We must differentiate between compositions of materials that are uniform (Tier 1.1), randomly mixed (Tier 1.2), ordered, clearly defined core-shell materials (Tier 1.3), and onion-like morphologies consisting of multiple shells (Tier 1.4). We must distinguish the chemical composition of the core from each of the shell(s). Note that the composition of the core may not influence surface reactivity in the case of core-shell NMs, although if the shell degrades over time, the core may be exposed. Ealia and Saravanakumar expanded classification and characterization of NMs by suggesting that optical, mechanical, magnetic, and electrical properties should be included to enable a prediction of their behavior in the different environments [43]. The particle color, optical density, visible and UV absorption, and reflection properties in solution or in a coating all provide to optical information. Mechanical properties include elasticity, ductility, tensile strength, and flexibility. Magnetic or electrical properties include conductivity and resistivity. Hydrophilicity, hydrophobicity, dispersibility, diffusivity, and sedimentation characteristics are also grouped here. The chemical properties impact on the reactivity of NMs with their environments. Factors affecting NM stability, such as moisture and heat are also included as classifiers, but are hard capture in its structural representation. Corrosive, anti-corrosive, oxidation, reduction, and flammable properties of NMs may also need to be considered as part of a wider set of characterization data. Despite the wide range of properties that may affect NM functionality, in practice only a subset of these are routinely characterized and reported. These mainly include chemical composition, size, size distribution, shape, crystallinity or chirality, and a minimal set of surface characteristics, determined by a wide range of physicochemical analysis techniques (Appendix A). 

Tier 2 represents the overall morphology of a given NM and encodes information on NM dimensions and shape (Figure 1). NMs were originally classified by Gleiter [44], as modified by Skorokhod [45]. However, the Gleiter scheme did not account for the diverse dimensions of nanostructures found in fullerenes, nanotubes or nanoflowers. Therefore, Pokropivny and Skorokhod reported a modified classification scheme distinguishing between 0 D, 1 D, 2 D, and 3 D materials [46]. It is based on the number of NM dimensions that lie outside the nanoscale (≤100 nm) range. Accordingly, 0 D NMs have all dimensions within the nanoscale (≤100 nm, nanoparticles or nanospheres), 1 D NMs have one dimension outside the nanoscale (nanotubes, nanorods, nanofibres, and nanowires), and 2 D NMs have two dimensions outside the nanoscale (graphene, nanofilms, nanolayers, and nanocoatings). 3 D NMs are more complex as they are distributed forms of nanoscale objects that are not confined to the nanoscale in any dimension (powders, dispersions of NMs, bundles of nanowires, nanotubes and multi-nanolayers, or nanoporous materials). 

ISO terminologies for nano-objects (ISO/TS 80004-2:2015) define nanoplates as having one dimension in the nanoscale and nanoribbons as having two larger dimensions that differ significantly. For nanofibers and nanoplates, one or more dimensions may or may not be in the nanoscale but must be significantly larger than the other dimensions. Definitions for nanostructured materials have also been published (ISO/TS 80004-4:2011) that extend to entities more complex these simple nano-objects. For example, a nanocomposite is a solid material containing at least one physically or chemically distinct region, or collection of regions, having at least one dimension in the nanoscale. Similarly, a nanofoam is defined as a liquid or solid matrix, filled with a gaseous phase, in which one of the two phases has dimensions in the nanoscale. A nanoporous material is defined as a solid material containing nanopores or cavities with dimensions on the nanoscale [47]. In medical applications, nanoporous materials may be filled with drugs, immune-modulating biologics, or disease modifiers as cargo to be delivered to sites of disease or injury to promote healing and tissue regeneration [48,49]. Note that the regulatory term *nanoforms* that emerged later (2019), is used to distinguish between compositionally similar NMs of different sizes, shapes, or coatings and is consistent with the ISO terminology for nano-objects. Size is highly relevant as it exerts a strong impact on NM behavior and function. It is a distribution not a single value, and multiple ways to report size have been used [50]. Electron microscopy is the method of choice for measuring size, also providing information on shapes and shape distributions of NMs. The morphology tier 2 thus contains information about the dimension (0 D–3 D), size and size distribution, and shape information from electron microscopy. For mesoporous 3 D materials, the average size and the size distribution of the pores must be defined. 

Size and shape determine the relative number of atoms at the surface and the surface properties of NMs. The surface-to-volume ratio can be modified by the changing size and shape of a nanostructured material. In very small particles, most atoms reside on the surface and have lower coordination numbers than those in the core. Novel material properties can thus result from surface defects in small particles. Vacancies can be the source of activity of oxide materials, e.g., in redox reactions. Moreover, the transport properties of ionic conductors originate from defects in the crystal lattice. Thus, there are new ways to tailor the physical and chemical properties of materials by controlling particle size, surface purity, or by selective interfacial doping [51]. 

Submicron-sized structures haves specific nanotopographies [52]. NMs with radii <10 nm have three main types of surface features: faces; edges; and amorphous surfaces. The atoms on edge sites may be more reactive than atoms of faces because their different chemical environments lead to charge redistributions and the lower coordination numbers compared to face atoms. Nanoframes and ultrathin nanowires are examples of atypical edge materials. Nanocrystals with face-centered cubic (fcc) structures and sizes ~1 nm have 100% of atoms on surfaces and edges, decreasing to 50% for 2 nm nanocrystals. The percentage of surface atoms decreases inversely with the and nanocrystal reactivity depends on its surface features, e.g., the number of steps, kinks, and terraces. NM geometries also affect their surface properties e.g., metal NMs with fcc structures have different proportions of atoms in the (100), (110) & (111 surface planes). NM surfaces can also have atoms with dangling or unsatisfied bonds. These atoms are subject to inwardly directed forces, so are closer to neighboring subsurface atoms than the interior atoms are to each other. Each face has a characteristic surface energy that depends on how many broken bonds are present at the surface. In metals with fcc structures, the (111) faces have three broken bonds, the (100) faces have four, and the (110) faces have five [53].

Tiers 3–5 (Figure 1) encode NM surface characteristics. Tier 3 describes physical and chemical surface parameters, such as solidity/roughness, charge, charge density, oxidation, or hydrophobicity (experimental octanol-water partition coefficients, challenging for NMs [54]). While some of these characteristics are valuable for grouping NMs and building predictive models, they *depend on external conditions*, such as the pH, solvent, or temperature of the measurement. A *NInChI* that encodes charge or hydrophobicity require metadata describing these conditions, which may be beyond its scope. Tier 4, surface functionalization, encodes the chemistries of coatings attached to the NM core and the type of linkage. It includes functionalization density, orientation, and binding type (reversible vs. irreversible), parameters difficult to characterize experimentally. Covalent linking results in irreversible surface functionalization with a known orientation. Non-covalent binding based on electrostatic, hydrophobic, hydrophilic, or van-der-Waals interactions is reversible and subject to external factors such as solvent polarity, ionic strength, molecular competition that will vary with environment. Tier 5 describes the nature of surface ligands—density, orientation, and distribution, currently poorly characterized but necessary to completely describe different NMs [55,56]. In nanomedicine, ligands are intentionally attached to NM surfaces for diagnostic and therapeutic purposes. These include bioactive substances and macromolecules such as proteins (or oligopeptides), nucleic acids (aptamers, derivatives), etc. Such ligands are usually bound irreversibly to the NM using covalent linkages. The chemical reaction used to couple them may be important. Ligands bound reversibly by physisorption are also used (e.g., in drug eluting materials) and are subject to dynamic exchange, a process to be captured as a NM transformation. 

Nanotopography can be important as it may affect the adsorption, orientation, and density of ligands. For example, molecular dynamics simulations characterized the adsorption behavior of branched polyethylenimine (br-PEI) on the surface of gold nanoparticles (Au NPs). Preferential adsorption of br-PEI onto the (111) surface of Au NPs was observed. Moreover, the br-PEI maintained a flat arrangement on the surface and stably wrapped the Au NPs, blocking the adsorption of water molecules and other free br-PEI molecules [57].

Several methods have been developed to monitor NM transformations in different media [34]. Templates for experimental workflows have been produced that capture relevant physicochemical properties, characterize transformations, document the exposure-relevant form, manage data traceability, and the minimal set of metadata required for different forms of the NM (i.e., tracking of its extrinsic properties modulated with the surrounding medium) [58]. Capture of (bio) chemical NM transformations by time and fate in different media is certainly beyond the capacity of current 1 D text representations. It is useful to distinguish between intrinsic and extrinsic NM properties, (charge, agglomeration, corona formation and exchange) that require metadata such as time and temperature [59]. Future developments in the RInChI may provide a framework for capturing transformed or aged NMs in the future.

In summary, the five tiers summarized in Figure 1 encompass issues most relevant to encoding physical, chemical, and biological properties of NMs. NMs exhibit a high level of structural and chemical complexity and diversity. Thus, consensus and compromise will be needed to capture the most relevant parts of it in an InChI-like string representation that follows similar rules as for small molecules, mixtures, polymers, and reactions. Here we propose the first specification for *NInChI* based other relevant extensions to InChI and an understanding of the complexity of the chemical and structural properties of NMs. We used case studies to test the capacity of *NInChI* to adequately represent different types and forms of NMs, and have incorporated information from stakeholders on how they would use the *NInChI.* We adopted a consensus construction approach, prioritizing essential, intrinsic NM properties, and using the minimal number of tiers necessary to describe diverse types of NMs. This allows filtering of an extensive list of potential features to a workable set that captures as much as possible of the NMs’ complex structure and chemistry. 

### 1.3. What We Intended the Case Studies to Teach Us

A 1 D line notation describing the structure of NMs, similar to SMILES or InChI notations for small molecules, could provide a simple, descriptive, and useful representation of NMs for literature and database searches and cheminformatics analysis. Such a machine-readable representation of entities can link NMs to biological datasets and physicochemical descriptors, enabling the building of information systems and NM property-biology models. It has already been demonstrated that a standard, compact, linear notation may allow some predictions [60], it may not distinguish between NMs with different surface ligands (case study 1), different symmetries or type of layering (e.g., carbon nanotubes of different kinds or graphene, case study 2), or with different structural types (e.g., amorphous silica vs. quartz for SiO_2_; rutile vs. anatase for TiO_2_), or chemical doping (case study 3). Encodings based purely on chemistry may be inadequate as they generate identical strings for materials with different properties and cannot encode complex 3 D structures. A chemistry-based description would, for instance, group together different nanoforms with the same core chemistry irrespective of shape or surface differences (see case study 1 and case study 6 on regulatory relevance of *NInChI*), and would fail to distinguish between a chemically doped and core-shell NM with the same overall composition (see case study 3). As we noted, graph-based NM representations are not useful because of structural complexity or because the NM structure is not known. 

Given the complexity of NMs, it is not surprising that there is currently no standard way to represent all their features and properties. A single linear notation will not completely resolve this issue. As we have stated, our goal is to create a universally applicable, widely supported notation capturing the most important features of the NM, augmented as necessary by additional metadata. To ensure utility, prospective *NInChI* stakeholder groups (database managers, nanoinformatics and modelling communities, and regulators) were consulted to understand their needs and applications for such an *NInChI*. To address as many of these as possible with a tractable encoding of NM properties, we used six case studies to validate putative *NInChI*s. Three case studies assess the suitability of *NInChI* for specific families or classes of NMs, and three look at implementation of the *NInChI* by end-users. End-users investigated how *NInChI* enhanced data management and FAIRness (case study 4), facilitated nanoinformatics research (case study 5), and supported regulation by identifying sets of nanoforms (case study 6). This identified the most important nano-specific features across case studies that must be captured in the proposed *NInChI.* This should ultimately result in a NM-specific simplified representation that: (i)distinguishes between NMs and groups similar NMs;(ii)enables extraction of the main characteristics of the NM directly from the representation. This is a machine-readable form usable by databases and literature-mining tools that can guide users to information on specific NMs that discriminates between bulk forms of materials or other NMs with other sizes, shapes, or surface modifications;(iii)allows users to merge relevant information from different sources, including data from other NMs with varying degrees of similarity to the material under study; and(iv)can be unequivocally decoded into structural information used to generate data-driven and physics-based computational models.

## 2. Materials and Methods 

The six case studies were conducted by multidisciplinary teams of expert experimentalists in nanosafety assessment, in silico modelers, data managers, and experts in regulatory issues and commercial nanosafety (data) service providers. Work commenced at a transatlantic workshop in Iceland in February 2020 and continued over several months via online discussion groups and regular virtual meetings. The workshop introduced the InChI and extensions for reactions (RInChI), mixtures (MInChI) and polymers (PInChI), that could be applicable to NInChI notation. Progress in materials modelling, metamodelling and nanoinformatics, and FAIRification of data were also reviewed. Initial suggestions for a systematic description of NMs included the proposal by Gentleman and Chan [61] for codifying NMs using 5 critical fields: chemical class (organic, inorganic and outer layer); size and shape; core chemistry (with constituents encoded by existing chemical conventions in alphabetical order); ligand chemistry (with functional groups ascribed to the core or shell); and dissolution kinetics (see Appendix A). The case studies showed that field 5 should be excluded from the *NInChI*, because NM dissolution is a behavior not a composition or structure property that is dependent on extrinsic factors (solvent, ionic strength, corona composition) and beyond the scope of the *NInChI*. Breakout groups identified key properties of NMs that impact their toxicity for specific NM families based on the first three case studies. These were: (i) presence of impurities; (ii) which atoms in cappings or coatings are chemically or physical bonded to the core and; (iii) the types of bonding in the core (ionic, covalent, or metallic crystals) and between the core and coating (electrostatic, disulfide, grafted, hydrophobic, etc.).

### 2.1. Experimental Case Studies

The experimental case studies drew on the work of Gentlemen & Chan and applied a bottom-up approach to determine the minimum number of descriptors needed. It determined what tiers may be borrowed or adapted from the reaction, polymer, mixture, etc. InChI extensions. 

#### 2.1.1. Case Study 1. Functionalized Gold NMs

This study employed a library of Au NMs of different sizes and surface ligands assessed for protein binding and cellular toxicity [62]. Gold is relatively inert and has an fcc structure with atoms located at the corners and the center of each face of constituent cubes. They have low dissolution rates under physiological conditions and are easy to functionalize by disulfide bonding to a range of small molecules or polymers, creating self-assembled monolayers [63]. They can be synthesized in different shapes and sizes, with excellent control over their size distribution. Thus, Au NMs are a useful starting point to understand what is needed to represent a simple NM. The library reported Walkey et al. consisted of 105 surface-modified Au NMs with 15, 30, or 60 nm cores, functionalized with 67 organic surface ligands. These included small molecules, polymers, peptides, surfactants, and lipids [62]. The ligands were neutral, anionic or cationic. Since small molecules already have InChIs (e.g., Cetrimonium bromide (CTAB)), this case study allowed us to explore combining the *NInChI* with one or more small molecule InChI codes. As Au NMs are usually supplied as aqueous dispersions, this case study also allowed us to decide whether the MInChI extension could be used to describe the wt% of Au NMs in dispersion. Thus, case study 1 focused on reporting size, size distribution, shape, and surface functionalization of the NM.

#### 2.1.2. Case Study 2. Graphene-Family NMs

This study used a library of carbon nanotubes (CNTs) from Sigma Aldrich to exemplify a range of different features. CNTs have additional features to those presented in case study 1. These include single or multiple walls, chirality (mirror images not superimposable), and the presence of defects and disordered regions that may affect their properties and toxicity [64,65]. They have multiple synthesis methods, some of which introduce substantial toxic impurities [66,67]. As with the Au NMs, CNTs can also be functionalized by small molecules and other ligands. Thus, case study 2 to considers new properties of chirality, defects, and impurities. 

#### 2.1.3. Case Study 3. Complex Engineered (Doped and Multi-Metallics) NMs

This study address issues associated with more complex chemistries and structures. Examples include chemically-labelled or doped NMs where atoms in the NM core are substituted by other atoms to facilitate tracking against a high background of core material [68], or to change the band-gap of the NMs and thus their toxicity [69,70]. Since the proportions of the two components in a doped NM are important, it assessed whether the MInChI extension could be applied to these solid solutions [71]. The ability to encode multi-metallics and to distinguish doped materials vs. core-shell NMs of similar composition was also explored. Thus, case study 3 focused on defining ordered vs. random mixtures and the description of multi-component core compositions, e.g., bimetallics.

### 2.2. Application Use Case Studies

These case studies probed how *NInChI* might be used in practice and identified the key features required by stakeholders (database owners, modelers and regulators). The aim was to a *NInChI* that would: facilitate integration and harmonization of NMs datasets and “FAIRification” of data; generate input structures for modelling (nanoinformatics) and; support regulatory activities such as grouping, read-across, and identification of nanoforms and sets of nanoforms. 

#### 2.2.1. Case Study 4. Encoding for Data FAIRness

This study 4 investigated NM-related data management and incorporation into the FAIR (*Findable*, *Accessible*, *Interoperable*, *Reusable*) data landscape: It determined the best ways to integrate and complement existing and evolving schemas for the use of persistent identifiers for NM compositions, formulations, batches, etc. and FAIR digital objects. RDF-based protocols are being developed across several FAIR Data Implementation Networks, including the recently launched AdvancedNano Integrating Network (coordinating data resources and stakeholders in the NanoEHS domain), to enhance the findability and interoperability of data across various domains. The case study tackled challenges identified in these networks that might may be addressed by the a *NInChI* and provided key examples of how the *NInChI* extends beyond existing identifiers, e.g., the JRC material identifiers, by providing more scientifically descriptive information about NMs.

#### 2.2.2. Case Study 5. Nanoinformatics Applications

*NInChI* in nanoinformatics was the focus of this case study. It considered how a *NInChI* will support both physics-based and data-driven nanoinformatics approaches. For physics-based materials modelling, we explored whether the *NInChI* could generate a starting structure for MD or quantum mechanics calculations. For data driven modelling, we investigated the use of NInChIs to train predictive models of NM physicochemical, hazard, or fate properties. Introduction of the first InChI revolutionized small molecule cheminformatics, enabling resources like ChΕΜBL to integrate data on chemicals from databases and the literature to produce a compound report card containing all available data about that substance. Our vision for the *NInChI* is that it will similarly allow the integration of nanosafety data from disparate studies on the same NM that will enable development of data-driven predictive models and workflows for meta analyses with a high degree of confidence. This case study thus focused on the utility of the proposed representation for materials modelling, meta-analysis, and nanoinformatics workflow automation.

#### 2.2.3. Case Study 6. NM Regulation

Regulatory challenges with *NInChI* are addressed in this case study. The introduction of nanoforms is the first step in accommodating different shapes, sizes, and coatings on a NM core that result in quite different fates and toxicities. However, the boundaries between different nanoforms and sets of nanoforms are not clear, a challenge identified by the experimental case studies. For instance, questions could be “what % doping is required to define a new nanoform, 1%, 5%, or 10%?” Similarly, “is 5% functionalization of an Au NM with a specific ligand equivalent to 10% surface coverage by the ligand?”, and “does the distribution of ligands over the surface matter?” Would all these materials get unique *NInChI*s or share a single *NInChI*? This case study aimed to address these issues. The enhanced discoverability and interoperability of datasets provided by *NInChI*s allows us to test different hypotheses and different methods of clustering nanoforms. This disclosed the situations where NMs count as a set of nanoforms encoded by the same *NInChI* and when a nanoform requires a unique *NInChI*. *NInChI* is potentially an important tool supporting read across and grouping and identification of unique nanoforms. Sets of nanoforms could potentially share a *NInChI* or an overarching family *NInChI*, with members being differentiated via an additional tag. This user case study thus explored the role of *NInChI* in NMs grouping, read across and similarity. 

### 2.3. Identifying Essential NInChI Features by an Iterative Prioritization Process

The six case studies above identified key *NInChI* requirements, both those common to all experimental case studies and those unique to specific NMs. Iteratively mapping of these to the user needs identified in the end-user case studies allowed prioritization of the essential list of NM-specific properties (see Section 3.7). Thus, we moved from wish lists from the experimental case studies, through a feasibility and needs analysis by the user case studies, to a preliminary specification of how a *NInChI* might look, shown schematically in Figure 2. The layers and their hierarchical ordering aim to provide an optimal balance between the needs of experimentalists, modelers, database owners and regulators for a well-defined yet sufficiently complete structural representation of NMs. Further refinements will be needed (e.g., merging or binning for those properties that exist as a continuum) based on stakeholder feedback, which this paper aims to stimulate, and as domain knowledge grows over time. Here we present an alpha version of the *NInChI*, as exemplified by the experimental case studies as proof of principle of its utility.

## 3. Results and Discussion

### 3.1. Case Study 1. Functionalized Gold NMs 

#### 3.1.1. Objective

This case study addressed the hierarchy of structural and chemical complexity shown in Figure 1 for different types of functionalized Au NMs. Figure 3 shows the properties of Au NMs considered—morphologies (nanoparticles, nanospheres, nanoshells, nanorods, nanoclusters, nanocages, or nanostars) and the chemical ligands used for surface functionalization. Different ways of reporting size and its distribution were addressed in this case study. The most relevant NM properties from this study may be implemented in different layers of *NInChI* to satisfy the information needs for other types of NMs. This case study focuses on all five Tiers in Figure 1 using a relatively inert but versatile NM with ranges of sizes, shapes and surface functionalities.

#### 3.1.2. Specific Features of Functionalized Au NMs

The level of detail needed to differentiate NMs and define their similarity may be application specific. Case study 1 identified that NM biological activity and physicochemical functionality are relevant. The ligand functionalized Au NMs from the Walkey paper [62] of three sizes required the following information to distinguish between library members: *Chemical composition.* The simplest NMs were spheroid Au cores with an organic surface ligand for stabilization. The size and shape of Au NPs were controlled by changing the synthesis conditions [72]. Chemical information is processed from the centre out, starting with basic information about core composition, size, and shape, before progressing to surface characteristics and functionalities, and even extending to interactions with surrounding molecules.
➢Thus, minimum information is the core composition and a representation of the stabilizing ligand.*Size*, *shape and morphology.* NMs are in general referred to by their core composition, with an associated physicochemical property and/or morphology, e.g., Au quantum dots, citrate-functionalized Au NMs, core-shell silver-gold NMs, etc. Their biological effects and physicochemical properties depend on the NM dimensions and shapes, e.g., spheres, rods, stars, cages, core-shell particles, etc. At the atomic level, a multitude of Au NM shapes can be differentiated [73]. Au nanospheres or colloids can be synthesized in an aqueous HAuCl_4_ solution using different reducing agents, e.g., citrate, which produces nearly monodisperse nanospheres [74]. The size of the nanospheres can be precisely controlled by varying the citrate/Au ratio, i.e., smaller amounts of citrate will yield larger nanospheres, and size variants differing by only few nm have been successfully synthesized [75,76]. Alternatively, Au nanorods are synthesized using templates, e.g., the electrochemical deposition of Au within the pores of nanoporous polycarbonate, or, alumina template membranes [77,78]. The pore diameter of the template membrane determines the diameter of the Au nanorod. The length of the nanorod can be controlled by the amount of Au deposited within the pores of the template [74]. Au nanostars have a thin, branch-like structure exhibiting plasmonic properties [79] and enhanced near-infrared light-absorbing capabilities, with reduced toxicity [80]. Octahedral solid core Au nanohexapods have been fabricated by reducing HAuCl_4_ with DMF in an aqueous solution containing Au octahedral seeds [81,82]. Case study 1 suggested that the core size, shape and nanotopography of the Au NMs (intrinsic properties) should be determined by direct imaging techniques such as scanning electron microscopy (SEM), transmission electron microscopy (TEM), and atomic force microscopy (AFM) [83] (see Appendix A) [84,85,86,87,88]. However, there are alternative ways to report size and shape and their distributions. Since Au NMs are not completely monodisperse, it is important to determine the particle size distribution to determine how agglomerate size affects toxicity [89], or to assess the quality of synthesized Au NPs [90]. Dynamic light scattering (DLS) is the most common sizing technique but has limitations, e.g., high polydispersity can distort the results [91]. Thus, consensus approaches to describing size distributions may be needed, e.g., DLS, SEM/TEM, field flow fractionation coupled to online sizing detectors, centrifugal techniques, nanoparticle tracking analysis and tunable resistive pulse sensing [50]. However, commonly the mean diameter is reported [84,92,93].
➢The structural representation of a NM must include size, size distribution, shape, and morphology distinguish one NM from another.*Dimension and thickness of coating or shell.* This property relates to chemical composition and morphology and may be difficult to determine. Examples include NMs where silica or polymer beads are coated with Au of variable thickness, creating Au nanoshells [94,95]. The diameter of the nanoshell is determined by the diameter of the underlying core, and the shell thickness is controlled by the amount of Au deposited on the surface of the core [74]. By varying the composition and dimensions of the chemical layers, nanoshells can be fabricated with surface plasmon resonance (SPR) peaks ranging from the visible to the near-infrared region, i.e., 700–900 nm [96]. Similarly, nanocages are hollow, porous Au NMs ranging in size from 10 to >150 nm. Silver nanostructures can be used as a sacrificial template and transformed into Au nanostructures with hollow interiors via galvanic replacement, e.g., a reaction between truncated silver nanocubes and aqueous HAuCl_4_ [96,97]. Au nanocages have been created with controllable pores on the surface [98]. The dimension and wall thickness of the nanocage is controlled by adjusting the molar ratio of silver to HAuCl_4_ [74]. Gold nanocages also be heated by light (photothermal effect) [82].
➢*These examples show that a shell formalization is needed that captures the dimensions and features of each shell in a sequential manner based on distance from the core*, i.e., *core-shell1-shell2* etc.*Au NM surface characteristics and functionalities*—Experimental determination of all relevant physicochemical NM surface characteristics such as roughness, charge density, oxidation, etc. is often infeasible so capturing these properties in a *NInChI* appears beyond the scope. This is less a limitation for intentionally synthesized conjugates. For example, PEGylation involves coating NMs by grafting, entrapping, adsorbing, or covalently binding to the NM surface to enhance its stabilization [99,100]. Covering Au NMs with polyethylene glycol (PEG) or its derivatives modifies binding of plasma proteins, interaction with opsonins, and clearance by the reticuloendothelial system [72]. Similarly, nanoflares are Au conjugates functionalized with oligonucleotide sequences complementary to a specific nucleic acid target (messenger RNA) hybridized to short sequences that fluoresce when bound to a target [101].
➢These examples indicate the need to describe organic coatings or biomolecules. However, quantitative and qualitative description of the coating entity (density, thickness, purity, orientation, bonding) may be beyond the scope of the NInChI, at least for now.*Au NM interactions with surrounding molecules*—The surface characteristics of Au NMs determine their life span and fate within the body and their toxicity [99,100,102,103]. As noted above, Au NP toxicity is affected by the type of particle coating [103], with polymer coatings increasing the stability and prolonging the NM circulation in the blood by reducing binding of opsonizing proteins [72]. Surface characteristics can influence the electrostatic and hydrophobic interactions between particles (i.e., agglomeration) and clearance by opsonization toxicity [99,100,102,103]. A protein corona can form on particles in vivo [104] that influences biodistribution, biokinetics and toxicity. An overview of toxicity studies of different Au NMs and their functionalizations is given in Appendix A which identifies the need for detailed structural representation of the ligands and surface functionalization.
➢Although relevant to the biological behavior of the NMs, protein corona and interactions with the environment are extrinsic properties or transformations that are beyond the scope of the proposed NInChI. They may, however, be suitable for a future extension by analogy with the RInChI.

#### 3.1.3. Conclusions on Relevant Features of a *NInChI*


Core composition and morphology, size and shape and their distributions, and surface functionalization by organic compounds or biomolecules are key properties needed to differentiate Au-based NMs. Surface functionalizing chemicals already have InChIs that can be added to the NM core *NInChI* or included as non-structural substances in the MInChI, although this would add further levels of complexity. These properties are also critical for their biological and functional performance. Moreover, reversible interactions with molecules in the environment add complexity, blurring the lines between intrinsic and extrinsic NM properties. Therefore, *NInChI* should: encode the major chemical component(s) and their ratio(s) if more than one chemical substance is present; categorize the morphology and quantify size and shape and their distributions; and encode surface modifications using existing InChI codes for the functionalizing components. As exemplified by Au-based NMs, capturing properties should satisfy these needs of other inorganic NMs (silver, silica, titania, different types of metal oxides), and potentially, well-defined biocompatible NMs used in nanomedical applications. Note that capturing dynamic behaviors (transformations) that depend on extrinsic factors (ions, proteins, or other available ligands), is beyond the scope of the of the current iteration *NInChI*, as are behaviours like dissolution. 

### 3.2. Case Study 2. Graphene-Family NMs 

#### 3.2.1. Objective

The first case study described the most relevant properties of inorganic NMs for implementation in different layers of *NInChI*. Chemical composition, size, and surface modification are also relevant for other types of NMs including organic. However, these materials, e.g., graphene-family NMs, introduce require new features beyond those presented in the first case study. We analyzed the selected group of organic NMs and identified additional properties important for implementation in *NInChI*. We focused particularly on chirality (also relevant to inorganic NMs), defects, impurities, number of layers (for nanotubes), and functionalization.

#### 3.2.2. Specific Features of Graphene-Family NMs

Here we focused on three important types of materials, graphene sheets, fullerenes, and carbon nanotubes (CNTs), and identified additional information needed to differentiate between family members. Figure 4 shows of additional features of these materials, including defects, end/edge functionalization, and chirality, that must be captured in structural representations.

*Graphene*. Graphene is the thinnest and strongest of the carbon allotrope NMs. Composed of a monolayer of sp^2^-hybridized carbon atoms, it owes its properties to the specific “honeycomb” crystal structure (Figure 4). This simple, planar structure has two basic edge structures, zigzag and armchair, [104,105,106] which influence the electronic and optical properties of graphene. Graphene structures can be converted into graphene nanoribbons (GNR), graphene oxide (GO), graphene nanopores (GNP), and graphene-nanoparticle (G-NP) hybrid materials. These can be functionalized by small organic molecules, polymers, biomolecules, NMs, and other carbon allotropes [106,107,108,109]. Thus, NM properties can be tuned to match requirements of specific applications. A recent study demonstrated the influence of impurities and defects. Mazanek et al. [110] compared the electrocatalytic activity of graphene with metallic impurities with ultrapure graphene. The impurities played a key role in the electrocatalytic activity, as the ultrapure graphene had much lower electrocatalytic activity than material synthesized according to standard methods. The production methods will impact on the presence of the defects [111]. These are disturbances of the lattice structure, such as point (e.g., Stone-Wales defect, single or multiple vacancies) or one-dimensional defects (e.g., dislocation-like defects, defective edges), which change the electronic, thermal, optical, and mechanical properties of the material [111]. Introduction of “bays” or protrusions into CNTs topology reduces their stability and can lead to changes in structure energy or cause destabilization (e.g., sheet bending) [112]. Not all defects are unintended. For example, intentional replacement of carbon atoms with boron or nitrogen atoms can tune the electronic structure of the graphene. 

➢
*Graphene introduces new structural features necessary to differentiate between members. These include edge-structures, impurities, and defects (Tiers 2 and 3 in Figure 1). MInChI may allow inclusion of impurities in a NInChI, while edges and defects could be defined as new categories. RInChI may facilitate grouping of synthesis-route specific properties if they are reproducible.*


*Fullerene*—Fullerenes are another carbon allotrope, of which the best-known example is C_60_. Its structure consists of sp^2^-hybridized carbon atoms arranged in sphere composed of 12 pentagons and 20 hexagons [113,114]. Larger fullerenes have also been isolated and characterized. They can act as hosts for atoms, ions and other small molecules that are encapsulated within the fullerene cavity (so-called endohedral fullerenes). C_60_ can be chemically functionalized as fullerene salts, exohedral adducts, open-cage fullerenes, quasi-fullerenes and heterofullerenes [114]. From a structural description viewpoint, these can be dealt with in a similar way to case study 1 e.g., they resemble hollow Au NMs (shells) and related structures. Given the very small size of fullerenes (<1 nm) it is not clear whether they are molecules or NMs. However, they are defined as NMs by regulators. 

➢
*These examples further elaborate formalization of hollow NMs representations.*


*Carbon nanotubes.* CNTs have many applications due to their unique structures and exceptional electronic, mechanical and thermal properties [114]. A single-walled carbon nanotube (SWNT) is a sheet of graphene that has been rolled up into a cylinder. It is composed of carbon atoms that form a network of strong sp^2^ C–C bonds. Graphene sheets, which can be rolled up to construct nanotubes with the zigzag, armchair or chiral structures. CNTs are characterized by the (n,m) notation for the chiral vector that determines two crystallographically equivalent lattice sites. The notation infers several nanotube characteristics indirectly: chirality; number of atoms; circumference; and diameter. All CNT have common attributes such as length, diameter, and open or closed ends (Figure 4). CNTs can also be built from more than one layer of graphene, denoted as multi-walled CNTs (MWNTs) [115,116]. To improve properties like catalytic activity, the surface chemistry of the CNTs can be modified by adsorption or functionalization. Adsorption of different molecules onto CNTs is achieved by electrostatic, **π**-stacking, hydrogen bonding, and van der Waals interactions. This changes the properties of CNTs without interfering with the graphitic structure. Chemical attachment of functional groups to the CNT sidewalls can change the structure of the CNTs, and modifies their physical properties. CNTs can be functionalized by small organic molecules, polymers, surfactants, metal NMs, biomolecules, etc. CNT structure can be influences=d by synthesis conditions; for example, carbon nanostructures can be doped with various heteroatoms where some carbon atoms in the layer are substituted by heteroatoms like Pt, Fe, B, or N [117,118]. Even a small change in the structure will affect the properties of the CNTs. 

➢
*As surface functionalization and doping by heteroatoms can alter the properties of CNTs, (see also case study 3) a mechanism to capture the bonding modality in a NInChI would be useful, (Tier 4 in Figure 1).*


#### 3.2.3. Conclusions on Relevant Features of a NInChI 

*NInChI* for the graphene-family of NMs must capture properties in common with the metal-based NMs discussed in case study 1 plus some additional properties. The starting point of *NInChI* for the graphene-family should be the dimensionality of the material (0 D fullerene, 1 D CNTs, 2 D graphene, Tier 2 in Figure 1). Depending on the specific entity, the structural representation should encode following attributes: *Graphene*. The proposed *NInChI* should indicate: the size of the graphene layer(s); the number of layers (single, bi-, tri-, n- layers); the topology of the structure if applicable (zig-zag, armchair); surface/edge functionalization and bonding mode; impurity information; and heteroatom doping. Although defects affect the properties of graphene, incorporation of information about them may be too difficult, especially for the non-intentional defects.*Fullerenes.* InChI notations for C_n_ (C_60_–C_90_) are already well established. However, to represent fullerenes as part of graphene-family materials and properly identify derivatives and surface modifications, *NInChI* could add as additional extensions that describe structural changes (i.e., the identity of surface functionalizations).*CNTs*. Graphene may be considered the parent material for CNTs, which are essentially folded (rolled) graphene sheets. Consequently, the *NInChI* will share common attributes with graphene sheet(s), such as: surface and edge functionalization; heteroatomic doping; and information on impurities. Additionally, information on the number of walls within the CNT should be provided (e.g., SWCNT, DWCBT or MWCNT). CNTs also require more extensive description of their surface and morphology. Therefore, the *NInChI* should be extended to include information on: the nanotube chirality defined by (n,m) notation (ideally each layer in the MWCNT should have its chirality defined); outer and inner diameters; length; surface charge; and specific surface area. CNTs exhibit additional, higher-level morphological properties to be included (to some extent) in the *NInChI*, e.g., the end-capping of nanotubes and their shape. End-capping could be partially covered by the edge functionalization parameter described above. However, this refers to the substances or heteroatoms bonded to the edges of CNTs, and should be distinguished from the outer carbon-capping of the tube structure. Due to the complexity of the capping parameter (studies show that the curvature of the cap can alter the properties of the CNT), only basic information on whether the nanotube is closed or open could be accommodated. Finally, information on the shape of the tubular structure should be included in the *NInChI* notation e.g., straight, branched, helical, waved, and more [119]. This could be accommodated by shape classes that group similar types of tubes.

The hierarchy nature of the *NInChI* could be easily adopted by graphene family materials without further modification. Tier 1 would include the type of NM (graphene, CNT, fullerene) and its pristine material characteristics (number of layers, presence of other heteroatoms), topology/chirality, and impurities. Tier 2 would include all morphological properties like size, length, shape, capping. Lastly, all information related to their surface properties and ligands is accommodated in Tiers 3 and 4 respectively (functionalization, bonded ligands).

### 3.3. Case Study 3. Complex Engineered (Doped and Multi-Metallics) NMs 

#### 3.3.1. Objective

Complex bi- or multi-component NMs are becoming common, further increasing the diversity of NM composition, structure, properties and potential applications [120,121]. Such NMs can be doped with inorganic [122] or organic dopants [123], and can be multi-metallic (≥2 metals) [124,125], providing an essentially infinite pool of compositional possibilities. Complex engineered NMs, while challenging to characterize and describe, exhibit very promising properties that can be fine-tuned to meet specialized needs. They are used for high-efficiency batteries [126], industrial surfaces (e.g., paints, surface coatings) [127], consumer products (e.g., cosmetics) [128], nanomedicines [129] and more.

Building on the previous two case studies, this case study identified specific properties needed to describe complex NMs. While most properties discussed in the previous case studies apply to complex NMs, there are specific additional characteristics that need to be included. In contrast to random mixtures already addressed by MInChI, complex NMs are well-defined mixtures whose structures are based on the structure of the ‘solvent’ (the major metal core structure of the NM) and changes induced by doping with another element. Bimetallic NMs can be alloys (i.e., homogeneous crystalline mixture of both metals), fused clusters, core-shell, or simply mixtures of monometallic NMs, shown schematically in Figure 5 [130]. Alloys can be predicted from the miscibility of metals.

#### 3.3.2. Specific Features of Complex Engineered NMs

Complex and doped NMs are solid solutions [131] having a defined structure, distinct from the random distributions of physical mixtures. Thus, a mixtures-like approach is suitable for this case, but without initially including any medium-related information. NM molecular descriptors could also be included, as they are structure-related and often correlate with biological effects. Solid solutions can be predicted by the Hume-Rothery rules [132,133] that account for the atomic properties of the metals involved. These include electronegativity, valency, and atomic radius, parameters that have not been widely used in NM analysis. Clearly, environmental (extrinsic) factors also play a role in the kinetics of the dissolution. Furthermore, while it has been shown that doping sometimes stabilize crystal structures [134], in general it induces instability that leads to higher dissolution and leaching of ions [135] that may be toxic [136]. NMs doping alters the band gap of the material compared to pristine NMs [122], which has been linked to oxidative stress in biological organisms when it overlaps with the redox potential for biological reactions [69,137].

An example of how the behaviour of NMs changes with crystal structure is TiO_2_, which can exist in anatase or rutile forms. The photoelectronic properties of TiO_2_ find multiple uses in industrial [138] or consumer [139] products. The photocatalytic activity of TiO_2_ can product reactive oxygen species (ROS) that are cyto- or genotoxicity [139]. This is especially the case for anatase NMs, due to their smaller crystal lattice size compared with rutile NMs, and its conduction band overlapping more strongly with the redox potential of biological reactions [69,137]. Some dopants enhance the photocatalytic properties of TiO_2_ NMs, while decreasing their toxic potential by shifting the band gap. For example, HfO_2_ shifts the TiO_2_ NM band gap towards lower values (out of the range of biological reactions) and the photocatalytic activity towards the visible part of the spectrum [122]. In contrast, doping of TiO_2_ NMs with Zr^4+^ produces the opposite effect, with higher toxicity compared to the undoped TiO_2_ NMs. 

➢
*These structures highlight a need to capture information on crystal structures, which may be mixtures of phases, and on amounts of dopants and distribution in the NMs. These features map to Tier 1 in Figure 1.*


#### 3.3.3. Conclusions on Relevant Features of a NInChI 

*NInChI* can be of very useful for describing complex multi-component NMs whose most relevant characteristics being the crystal structure of the NMs. Thus, an inside-to-outside description of the NM core chemical and tunable molecular characteristics is most appropriate. Consistent with the tiered approach (Figure 1), Tier 1 should contain information on the core chemistry (metals/elements present including any organic or inorganic dopants), the doping or metals ratio and some characteristic molecular properties, defining their behaviour, such as absolute electronegativity, band gap etc. Furthermore, information on the NM’s crystal structure and deviation from that of the pure material should also be included (e.g., crystal structure, space group, unit cell dimensions and strain compared to the original undoped structure) and whether the NM exhibits a layered or biphasic structure (e.g., core versus shell chemistry, or fused cluster as per Figure 5). Tier 2 should encode macroscopic, morphological and physicochemical properties of the NMs, similar to the previous case studies. These include the NM morphology and shape (e.g., diameter, long axis, short axis, side). The shape, aspect ratio, and size distribution should be reported to identify deviations from the perfect geometrical shape and to categorize the NMs according to whether their behaviour is closer to that of the bulk (>30 nm, approaching bulk material in terms of quantum properties), nanoscale (10–30 nm), or quantum matter (<10 nm e.g., quantum dots). Tier 3 can describe the NM surface characteristics such as coating or capping, the type of material used, and any surface functionalization. Finally, the tier 4 (if needed) may contain information on experimental/storage conditions (NM provenance) such as ageing and deviations from the original complex structure arising from this.

Complex NMs require structural representation in terms of experimental characterization and metadata that results in a greater need for data interoperability based on the FAIR principles (see case study 1). FAIR principles require clear and unique identifiers, adequate information on the material, and support for merging of disparate datasets for meta-analyses and predictive nanoinformatics models for NMs properties and behaviour. 

### 3.4. Case Study 4. Encoding for Data FAIRness 

#### 3.4.1. Objective

As described in the first three case studies, there are many types of NMs, for which large amounts of data/metadata are available that are used diverse purposes. The two case studies that follow this one probed data and structural representation usage in nanoinformatics (case study 5) and in regulation (case study 6). It is essential that the data produced and integrated from different sources is curated, quality-controlled, and stored in a way that allows it to be shared widely to generate new knowledge. Nanosafety research has triggered innovations in data management, ontologies, mapping, and curation methods. The process of defining requirements for annotation of datasets with rich metadata, as described in another paper in this special issue [58], is underway. The objective of this case study is to describe typical use cases, from curation and management to search and retrieval, and to abide by the constraints that the FAIR principles impose on structural, machine readable unique identifiers. “FAIRification” is a technical process, generally assigned to database owners rather than dataset owners.

#### 3.4.2. Specific Use Cases—Nano-Related Data Management, Analysis and “FAIRification”

This case study highlighted the use of *NInChI* to enhance nano-related data management and analysis, including its incorporation into the FAIR data landscape. Specifically, we assessed how well *NInChI*s support “FAIRification” of NMs datasets and supplement previously developed persistent identifiers, support the Findability and Interoperability of datasets, and enhance their Re-usability. It exemplified how *NInChI*s aid organization and discoverability of data using a subset of particles with one NM composition.

*Improving FAIRness using notations encoding details of specific NMs characteristics*. Linear representations are needed for search engines to index and resolving researches, as exemplified by the DOIs used for scientific publications. *NInChI* would allow indexing of NMs used in projects, publications, datasets etc. making them *findable* [140]. It would allow us to search for NMs with specific properties, such as potential toxicity. Chemical descriptions are inadequate (because different forms of the same material have different properties) or incomplete (due to the inability to encode complex structures). These descriptions would group nanoforms solely on core chemistry irrespective of shape or surface differences (see case study 1) or would fail to distinguish a chemically doped from a core-shell NM of equal composition overall (see case study 3). Representations based only on chemical composition only allow us to find data about all NMs of a specific composition regardless of size, shape, coating, crystal phase, functionalization etc. Indeed, the JRC representative industrial NM identifiers perform this role for the sub-set of 33 NMs used in the OECD sponsorship programme and related activities [141]. Thus, a search query using JRCNM01001a (a TiO_2_ NM) will identify publications using this particular NM, but these identifiers only cover a tiny fraction of available NMs. The Appendix A includes an example of the range of nanosafety literature utilizing JRCNM01101a (a 152 nm ZnO NM) based on a Bioschemas search (Appendix A). Including additional structural information in the *NInChI* will allow more specific searching, and identification of specific NMs with properties included in the representation. The physicochemical properties included in the layers of the new *NInChI* notation will make data more *findable* and enhance *scientific interoperability* by allowing exclusion of NMs that meet some but not all of the criteria contained in the representation (e.g., only TiO_2_ NMs that have at least 80% anatase and a specific surface functionalization). This capability is available in the InChI used to find information about similar chemicals [142]. If the *NInChI* includes the InChIs for the ligands used to functionalize the surface, then chemically searches can use it to identify data associated with this ligand. 

➢*NInChI will enhance FAIRness of NMs datasets by providing a higher level of indexing based on information in the different layers. Specific NMs can be found based on size, shape, surface coating* etc. *as encoded in the NInChI. Similarly, scientific interoperability of datasets will be enhanced by exclusion of non-relevant datasets.*

*NMs batch-to-batch variability*, *ageing*, *and similarity of NMs for dataset integration.* Synthesis of NMs is often a stochastic process that results in considerable batch-to-batch variability [143]. Different batches of NMs may have slightly different properties that induce different toxicity. The NanoCommons Knowledge Base assigns a unique identifier to each batch of a specific NM, even if the only measured difference is the concentration of the NMs in suspension. Accordingly, measurement data is associated with each production batch, allowing subsequent comparison of batch physicochemical characteristics and their toxicities to identify potential correlations or differences. Users of the data can then choose to group the NMs as similar or not. However, clear measures of equivalence have yet to be defined that would allow automatic integration of information from different batches. Similarly, the dynamic nature of NMs and their propensity for transformation leads to changes in NMs physicochemical properties during storage [59] or when released into the environment [34]. However, NM transformations are inherently linked to the physicochemical properties of the pristine NM. For reporting and database purposes, an aged NM should be linked to its parent pristine NM and to temporal characterization data. Within the NanoCommons Knowledge Base, pristine and aged NMs carry different identifiers, effectively treating them as separate production batches. Part of NanoCommons support to the H2020 NanoFASE project focused on extensively characterizing the transformations of NMs in the environment. The concept of a NM “instance”, reflecting the NMs characteristics at a specific time point in an experiment under a specific set of environmental conditions, was introduced (adapted from the US CEINT Centre’s NIKC database). Each NM instance receives a separate identifier that distinguishes NMs retrieved from different environments (e.g., stock solution, in exposure media, after incubation with test organisms for some period, etc.). The linked set of NM instance IDs can then be integrated with complete datasets, and properties or changes to the NM imposed by environmental interactions or storage can be identified, either from the metadata describing particle fate or via the NM instance IDs. While beyond our scope of the alpha version of a *NInChI*, we note that concepts from RInChI that relate reactions as transformations may allow for environmentally transformed NMs to have a *NInChI* that is linked to, or is an extension of the pristine as manufactured NM’s *NInChI*. Methods to distinguish whether two NMs are identical or similar for matching during data integration are required, especially for comparing assay or exposure data across datasets and production batches for example. However, it is not clear which physicochemical properties must match, and complexities associated with different measurement techniques, experimental protocols, and metadata standards (including batch identifiers), make identifying and matching NMs a challenge [144,145]. Can a *NInChI* provide enough information on the composition of the NM to quickly estimate similarity? By analogy to unique entity references used in molecular biology (e.g., NCBI Gene or Genbank gene IDs or UniProt Protein IDs), similarity should relate to the measured entity, not the measurement method. For example, the value of gene expression GeneID: 1956 will vary widely depending on the context of the measurement, such as the measurement device and the experimental design. Any context dependent variation is separate from the measured object, the gene, by agreement that a DNA sequence of sufficient similarity will be accepted as being the same gene. The context described in the metadata must include enough information to ensure that measurements are reproducible by others. However, in contrast to gene sequences, NMs are not precisely defined entities, rather somewhat heterogeneous distributions. The properties of NMs are important for defining a NM yet these properties may vary depending on the measurement device used and the specific protocol applied. For example, it is clear that size included in the *NInChI* will not be exact but will already provide a grouping of sorts. We will not define size in the *NInChI* as 2.2342 nm but as 2 nm. Thus, while different techniques for measuring size (e.g., Appendix A) give different values they will be broadly in the same range. If they differ significantly, it is because they are measuring different endpoints, e.g., core size by TEM vs. hydrodynamic size by DLS. Thus, for data management *NInChIs* should specify the type of size (core, hydrodynamic) to ensure consistency and comparability of datasets and facilitate automated identification of similar NMs. 

➢*If a NInChI is used to establish similarity of NMs (across batches, following storage or ageing*, etc.*) its representation should include additional methods for determining the relevant properties to ensure direct comparability/interoperability. We therefore argue for a NInChI encoding sufficient information to quickly gauge similarity that is adequate for most applications that integrates measurements from multiple batches and samples of NMs.*

*Nano-related data management and analysis: integration of computational NMs.* The European Registry of Materials was established to allow researchers to register NMs they are working on at an early point in their data generation process [146]. It supports embedding of data management practices into nanosafety assessment and encourages the research community to be consistent in their naming and referencing of specific NMs in their publications. It ensures that the identity of a given material can be established even before any properties are known (measured) and encourages data management earlier in the data generation cycle, and ensures data are *findable*. However, each project or lab working with a specific NM registers their NM batch, highlighting the need for clear principles for assessing similarity of NMs that allow integration of datasets for the same NM, as noted in the previous use case. *NInChI* will enable researchers to generate a structural representation as soon as they have the idea for a NM (it does not even have to exist as yet, as long as it can be drawn and obeys the laws of chemistry), facilitating indexing of NMs from their point of conception. As the community moves towards in silico approaches for nanosafety assessment, more and more virtual NMs may be generated using approaches such as molecular dynamics [147] or other materials modelling approaches. While we have not specifically included an indicator for computationally derived NMs it in the current proposal, the *NInChI* could easily include one to allow integration of experimental and computational data and provide transparency on the origin of the NMs. This will be particularly relevant as nanoinformatics models evolve in complexity and are interlinked, to ensure that circular arguments are avoided wherein in silico NMs are used to generate part of a model and then used to validate another part of a model.

➢*NInChIs can support the integration of computational NMs and their associated simulation datasets into nanosafety databases, with transparency around the origin of NMs datasets (experimental versus computational), through inclusion of notation to indicate* in silico *NMs.*

#### 3.4.3. Conclusions on Relevant Features of a NInChI

A fully developed *NInChI* will be valuable for nano-related data management and ongoing FAIRification of nano and advanced materials data resources. By defining a consistent identification scheme for NMs and by harmonizing the *NInChI* with the FAIR digital object framework, it will be possible to find, mine, and process data according to any specific NM search query (as long as it is encoded in the structural representation). The *NInChI* adds granularity to the information when searching, analyzing, and comparing the physicochemical properties of NMs This facilitates modelling of specific properties of the NM, not restricted to comparisons of approximately similar materials. Whether it is a reference material, an industrial product, a material of regulatory or research interest, or purely a theoretical or computational NM, the ability to search and use data related to its intrinsic parameters will increase our ability to integrate datasets, and to analyze them to discover which physicochemical parameters are most relevant to the property under investigation. This will be particularly important for grouping and read-across studies, safe-by-design, and the design and prediction of NM performance in product or process development. Future extensions incorporating concepts from RInChI for transformations may allow linking of temporal datasets in a meaningful manner. This will allow NMs aged during storage, transformed by the environment or upon contact with living systems to be linked to the pristine or as-manufactured NM characteristics.

### 3.5. Case Study 5. Nanoinformatics Applications

#### 3.5.1. Objective

Due to the increasing number of engineered NMs entering the market, it is becoming intractable to assess their safety experimentally using standard tiered in vitro and in vivo approaches. Ensuring safety, reducing the burden of testing, not unduly stifling innovation or commercial use, and the growing body of nanosafety data has stimulated research into computational modelling and in silico risk-assessment approaches [7,148]. The appeal of in silico approaches is their time and cost efficiency in comparison to experimental methods but is also due inherent problems of the latter. In vitro methods have limited predictive power for human toxicity and animal experiments have increasing ethical issues. In silico methods can combine data from multiple experiments and enable generation of read-across models derived from data-rich similar NMs, partly guided by in vitro experiments (chemical-biological read-across). The adverse outcome pathway (AOP) framework, for instance, represents an integrated approach to testing and assessing (IATA) safety of NMs [149]. To enable in silico screening and regulation by grouping, NM structure must be encoded in a systematic manner that reflect the degree of similarity between the NMs and allow datasets to be integrated and compared with confidence.

In the past decade, a variety of data driven computational approaches, such as Quantitative Structure-Activity Relationship (QSAR) models and read-across approaches that use diverse machine learning methods, have been used to predict NM-related toxicity [10,148]. The goal of these methods is to map the material description and intrinsic/extrinsic physicochemical properties to the biological outcomes to identify NM properties of concern, and facilitate design of NM that avoid, reduce or modulate these properties [150,151]. Computational techniques provide additional benefits where specific descriptors or properties are not readily measurable, as they can derive these using chemistry- and physics-based materials modelling [148]. This is particularly important for virtual NM not yet synthesized, where prediction of their properties is important. An overview of various computational models for NMs is given in recent reviews [10,150,151].

The availability of structured, systematic, and machine-readable data is the crucial enabling step for advanced analysis of NM properties and construction of predictive models. Data driven machine learning is dependent on sufficient amounts of reliable, well-structured data to train models. Data quantity, quality, and diversity are major requirements for any machine learning modelling task, thus any improvement in the collection, curation, and organization of data and metadata is of interest to the nanoinformatics community. As discussed in the previous case study, data quality control, management, “FAIRIfication”, and retrieval, are essential for nanoinformatics workflows, and nanoinformatics will benefit from a viable *NInChI*. Successful implementation of the *NInChI* will likely uncover other challenges within the nanoinformatics framework to be addressed. Two major ongoing tasks within nanoinformatics are generation of more effective NM descriptors and read-across from data-rich NMs to data-poor NMs.

#### 3.5.2. Specific Use Cases—Calculating Nanodescriptors and Enabling Read-Across

Computational NM characterization can generate detailed, quantitative information on physicochemical properties of NMs, providing potentially valuable insights into the toxicity mechanisms. They can be used to interpolate experimental datasets or predict properties for new or untested NMs. Grouping and read-across methods use similarity of NMs to predict the unknown properties of materials, thus allowing the community to efficiently screen novel materials for the specific activities. As noted in case study 4, definitions of NM similarity and criteria for establishing similarity for automatic integration of datasets are needed, a task *NInChI* will support. Here, the use of *NInChI* to generate virtual NMs, for grouping NMs, and for NM read-across is explored.

*Calculating NM descriptors based on the NInChI.* A key step in building predictive models for NM properties is extraction of information about NM physicochemical characteristics, such as the chemistry of the core, the size, crystal polyform, and shape of the NM under study. Such information is currently extracted from crystallographic information files or compiled from the data and metadata for a specific particle. Simulations at different levels of complexity can be performed using this information. A *NInChI* could significantly contribute to automatic extraction of input data for computational modelling, as its layered approach could supply information for initiating and executing the simulations at different spatial resolutions within a multiscale modelling approach. For example, to perform atomistic simulation of NMs using modelling tools developed within the NanoSolveIT project [149] information is required about NM chemistry, nanoparticle core size and shape, as shown in Table 1. Based on these input parameters, multiple advanced characteristics of NMs can be derived, such as structural and energetic descriptors (total number of atoms, number of surface atoms, surface energy, electronic structure descriptors, heat of immersion in water, etc.).

➢
*NInChI supply information for simulation, and link input and output data and simulation parameters as part of an automated nanoinformatics workflow. All model data can be stored and linked together with the corresponding NInChI. This allows data retrieval based on specific NM queries. Simulation parameters will also be stored as meta-data for a given NM and modelling method (Figure 6). An example of the three components for the nanodescriptors calculations based on atomistic simulations is given in Table 2.*


*NMs similarity searches based on computed descriptors derived from NInChI.* Nanodescriptors can be generated from the *NInChI*. This can enable searches for quantitative correlations between the calculated nanodescriptors and a property or activity of interest. Nanodescriptors based on the *NInChI* can also serve as fingerprints identifying similar NMs and be used within a read-across context. The k-nearest neighbor (kNN) algorithm, implemented within the Enalos Cloud Platform has been successfully used for read-across [152,153], and a similar approach could be used for the information encoded in the *NInChI.* The layered approach of the *NInChI* enables searches for similarity at different levels of complexity based on *NInChI* layers, while the content of the *NInChI* layers can be used to define similarity measures.

➢
*NM information encoded in the NInChI will enable grouping of NMs based on compositional and structural similarity, and enable generation of computational nanodescriptors to encode the whole NM.*


#### 3.5.3. Conclusions on Relevant Features of a NInChI

The hierarchical representation based on standardized measurements of NM physicochemical properties enables integration of nanosafety data from disparate studies for the same NM. It also aids design and development of NMs and advanced material formulations on a component-by-component basis. Bioinformatics has demonstrated the advantages this approach. Advances in biopharmaceuticals, health research, and basic science are being driven by advanced computational tools predicated on the bioinformatics space being searchable and interoperable. A *NInChI* would aid in NM construction, performance-testing, modelling, grouping, and read-across. It would help unlock better understanding as to which NM components are most responsible for their impacts on human or environmental health. Such a system is therefore a key requirement in the future development of nanoinformatics.

### 3.6. Case Study 6. NM Regulation

#### 3.6.1. Objective

As with other case studies, we need to set the scene on representations of materials used in regulatory settings, especially within the regulatory IUCLID (International Uniform ChemicaL Information Database) chemical data management platform (https://iuclid6.echa.europa.eu/) [154] that is endorsed by the European Chemicals Agency (ECHA) and soon also by the European Food Safety Authority (EFSA) for preparing and submitting dossiers. Currently, CAS registry numbers (RNs) are mainly used, but this results in inconsistencies even for small molecules. This is due to the somewhat unsystematic way substances are registrated at the Chemical Abstract Service [155] and their use in databases, the literature, and the regulatory system: RNs are often assigned purely on the basis of the chemical structure, which often leads to ambiguities due to multiple substances with different RNs corresponding to the same chemical structure. This is similar to the multiple to-be-registered substances that correspond to one reference substance defined in IUCLID6 [154].For NMs, there is the additional, more severe problem that no RNs exist for the different nanoforms, defined in EU chemicals legislation (Registration, Evaluation, Authorisation and Restriction of Chemicals or REACH) as NMs with the same core chemistry but different sizes, shapes, coatings. For these it is suggested that the bulk or micron-sized materials RN be used. ECHA are currently addressing this in IUCLID6 by specifying different *assessment entities* within a single substance registration.Commonly encountered mixtures of known or unknown composition and even whole classes of molecules like classes of enzymes receive a single RN.

As RNs are designed to designate only one substance, the first two examples above are clear violations based on a strict separation of the meaning of the term *substance* from the term *compound* supported by us and also by major databases like PubChem.

According to this definition, a compound is a pure (perhaps virtual) chemical that can be represented by a single chemical structure, regardless of how complex. In contrast, a substance is a real-world object usually consisting of one single main compound with different impurities or additives, but can also be mixtures or solutions. Thus, since 100% pure chemicals are impossible, all substances consist of several molecular structures. In REACH this is addressed by regulating substances but having reference substances associated with them, usually the main component. We would argue that this reference substance should be called the reference compound, since it is not representing the actual substance registered but an idealized state of a pure compound. For NMs, the current implementation of a reference fails because it is not possible to represent it in sufficient detail to discriminate between bulk and nano-structured forms using only identifiers like RNs or molecular representations like SMILEs or InChIs. Thus, these references are mainly used to group similar substances (different nanoforms are regulated independently because they do not behave similar to the bulk material) and to determine which regulations should be applied. The concept of nanoforms potentially addresses this issue but was introduced but without any guidance on how a nanoforms should be represented. InChIs as structure representation can complement the RN substance identifier for small molecules to assess similarity. Similarly, a representation of substance and reference structure representations for NMs must also be incorporated into IUCLID, as we make progress on identifying unique nanoforms. The *NInChI* proposed here covers compound representation, and the European Repository for Materials (ERM) identifier is another possible substance identifier.

#### 3.6.2. Specific Use Cases—Nanoforms Concept and NInChI as Solution for Regulators

One of the most important applications of identifiers/materials representations is in regulation. Surprisingly, regulatory systems currently use the non-unique RN as their main identifier. Indeed, as recently as August 2020, the curators of the US EPA’s Distributed Structure-Searchable Toxicity (DSSTox) Database suggested that substance registries like RN are a vital bridge enabling communication between frontiers of chemistry not currently well-supported by InChIs. This may be true at least until technological advances in chemical storage and canonicalization enable the InChI to serve as an effective identifier for all compounds of interest [156], although RNs have the serious limitations discussed above. Here we provide a first step towards implementation of an InChI for NMs that addresses these challenges in a more systematic way: *Fitting the needs of the nanoforms concept of REACH and ECHA*. According to REACH Regulation (Annex R.6-1, [40]), a *nanoform* is a form of a natural or manufactured substance containing particles, in an unbound state or as an aggregate or agglomerate where, in which 50% or more of the particles have one or more dimensions in the range 1–100 nm. This includes fullerenes, graphene flakes and single wall carbon nanotubes with one or more external dimensions below 1 nm. A *set of similar nanoforms* is a group of nanoforms for which hazard, exposure, and risk assessment can be performed jointly. A justification must be provided to show variations within these boundaries does not significantly affect hazard, exposure, and risk assessment of the similar nanoforms in the set. A nanoform can only belong to one set of similar nanoforms. ECHA developed a stepwise approach following the steps outlined by the OECD guidance on grouping of chemicals [156]. ECHA requires information on the nanoform that includes composition of the substance, impurities or additives, surface treatment and functionalization (chemical coating and surface treatment(s) applied to the particles). It also includes physical parameters such as size distribution, shape aspect ratio and other morphological characterization data, crystallinity, information on assembly structure including (e.g., shell-like structures or hollow structures, if appropriate), and specific surface area (e.g., porosity). The user has to measure each property (using a standardized protocol), report the method and the results in IUCLID [157]. A set of nanoforms can exhibit a range of values for each property provided that the range does not impact the nanoform’s risk. The regulation requires that at least 50% of particles (number distribution) is within the range of 1 to 100 nm, with further information on the particle size distribution (e.g., d10, d50, d90 values). The registrants must define the boundary defining the set of similar nanoforms, for example by specifying the minimum d10 and maximum d90. A set of nanoforms should exhibit similar dissolution rates, toxicokinetic behaviors, fate and bioavailability and ecotoxicological parameters.
➢By using a structural representation, rather than a chemistry-unaware substance identifier, in Tiers 1-3 (Figure 1) encoding composition, size/shape and surface coating of nanoforms, NInChI will support the differentiation of individual nanoforms (assessment entities) independent of their inclusion in identifier repositories controlled by third-party organizations like the CAS registry.
*NInChI as a solution for the regulatory needs*—*Moving from CAS to NInChI*. CAS RNs are completely arbitrary, contain no intrinsic information, and are easily misused as there is no way to validate them other than locating them in a database. Case study 4 highlighted some of the challenges presented by NMs regarding identification and linking of datasets to the specific NMs, batch-to-batch variability, NM ageing and transformation, and integrating data sets with confidence in the absence of unambiguous identifiers. Thus, for NMs there is a clear need for a representation like *NInChI* to replace or augment the CAS RN, or other chemistry-unaware identifiers, to increase confidence in datasets used in weight of evidence, grouping, read across, and QSAR modelling approaches. Using a *NInChI* to validate datasets, by checking the consistency of the data for the object under investigation, would significantly boost the quality of QSAR and read-across models, and the confidence with which they can be applied in nanosafety. Thus, in terms of addressing regulatory needs, *NInChI* will enable:
a.Distinguishing of different nanoforms for registration (importance of standardized identifiers and structural representations, unified data management processes, etc.). The tiers in Figure 1 have been mapped to the information requirements included in Annex R.6-1, including shape, size and surface coating considerations and, as such, *NInChI*s will be an important means to differentiate individual nanoforms. By adapting the MInChI extension that includes ranges, we envisage incorporating a set of nanoforms into a single *NInChI* by providing ranges within which specific NM properties can vary while their toxicities remain the same, thus providing boundaries for a set of NMs. While this is not included explicitly in the current examples, an extension in a subsequent iteration of the *NInChI* would be possible.
➢*NMs information included in the NInChI will support grouping of NMs based on both compositional and structural* properties.
b.Read across and grouping using *NInChI* for predictions, as described in detail in case study 5. While QSARs are well established for small molecules, their acceptance for regulatory purposes is still limited, mainly due to dataset uncertainty and often poor documentation of models in the QSAR model report forms (QMRFs). *NInChI*s will make it straightforward to update existing QMRFs with the *NInChI*s of all NMs that were used as part of the training and test sets. Using the *NInChI*s will enable extraction of NM structural information from databases and visual presentation. This will aid expert evaluation and interpretation of QSAR models for grouping strategies, determine whether it is applicable to the NMs under evaluation, and allow independent assessment of predictions and structural similarity. The notation that describes NMs from the center outwards will also allow a simplified graphical representation of the NMs key elements, potentially as a 2 D representation of the 3 D structure, as shown schematically in Figure 7.
➢NM information included in the NInChI supports verification and validation of grouping hypotheses based on simplified visualization of chemical and structural information.

#### 3.6.3. Conclusions on Relevant Features of a NInChI

The proposed *NInChI* would be beneficial for the regulation of NMs, and advanced materials more generally, as it allows for a more precise encoding of the components of what are increasingly more complex materials. It will allow more accurate descriptions of nanoforms and more valid comparisons of materials and their component layers, leading to improved risk assessments. Regulation of NMs is particularly challenging because of the variation in assays used to determine whether materials are adversely affecting human or environmental health.

### 3.7. NInChI—A Proposal for a Layered Approach for Uniquely Identifying NMs

The case studies demonstrate that experimental, nanoinformatics and regulatory groups have a significant overlap in expectations and requirements as to the most important features that the *NInChI* must encode. These requirements range from a simple, unique identifier to an almost complete description of a specific complex NM. It is clear that not everything can be accommodated into a single linear notation that is generally applicable to all NMs, easy to use, unique for each material, useful for grouping materials into similarity classes, but also sufficiently information-rich to satisfy the needs of data-driven and physics-based materials models. Therefore, the first step in generating an alpha version of a *NInChI* was to filter the NM properties listed in the case studies and categorize them into must have, nice to have, and beyond the scope of *NInChI* (see Table 3). Even if there were no limitations in performing the case studies, it is clear that the *NInChI* is meant to be a structural representation like the original InChI and not a complete characterization of the NM. Therefore, many extrinsic physicochemical, exposure, and hazard properties which depend on the NM’s environment are outside the scope of the *NInChI* even though they are important for the toxicity and the risk assessment of NMs. However, these properties are dependent on the structural features for which experimental values can be found in databases or, ideally, can be determined using nanoinformatics approaches based on the *NInChI*. Thus, categories 1 and 2 are essential to find data, generate input for in silico approaches, and to separate nanoforms in the regulatory setting. The additional data and the associated metadata provided by the database, computation, and regulatory dossier then transforms the *NInChI* into a real-world or virtual object used in experimental or computational studies. Category 2a differs from 1 by inclusion of properties that are either hard to determine or specific for one type of NMs and would not be available for the majority of NMs. Category 2b lists some extrinsic properties that make the representation more specific to NMs in specific conditions compared to the more general parent class or the pristine reference material. Even if the properties in category 2 in the proposed *NInChI* are not included initially, they may be included in later versions. It might be necessary to include additional metadata extracted from relevant studies to understand the conditions in which this NM form exists, and how the properties were measured (applies to category 1, see below). Category 3, encodes NM properties retrieved from databases, predicted, or reported in dossiers, not for discrimination purposes but to fully characterize and perform risk assessments using integrated approaches. Using these criteria, we propose the groupings in Table 3, to be discussed, refined and extended subsequently by the user community.

*NInChI* are primarily designed to be machine-readable. Adopting the layered structure of chemical InChI was identified as the most promising approach to developing a useful NInChI that is the best compromise for all stakeholder groups. In this way, individual subparts can be included in the notation to group information by a specific aspect, which can optionally cover properties only relevant for specific NMs or specific use cases. Other technical requirements are that: It should be a unique representation, where a specific NM is always represented by the same *NInChI* and a *NInChI* is always associated with the same NM or group of closely related NMs. The latter modifies the one-to-one relationship of InChIs described in the introduction by accepting the stochastic nature of many NMs, discussed in more detail below.The structure should be optimal for extraction of specific information by a computer but also by a trained person (i.e., should have a degree of human interpretability)It be compatible with other notations in the InChI universe, reusing concepts or extensions of these notations or even incorporating their complete representations as part of the *NInChI*.

The PInChI, MInChI and RInChI went through similar design cycles and showed that the notation layers do not necessarily have to correspond to universal concepts in the minds of researchers. For example, chemists clearly distinguish between reactants and the products with the exception of equilibria. However, the RInChI specifies these chemicals in layers 2 and 3 and defines which of these layers are reactants and products in layer 5. Additionally, distinct non-structural substances (i.e., enzymes, natural substances, heterogeneous metals, etc.) that are included in layers 2, 3 and 4 are specified in layer 6. Similarly, the hierarchy of tiers for NM characterization in Section 1.2 is based on the consensus of the authors representing different stakeholder groups, and was used to guide the case studies and the design of the *NInChI*. However, there is no need to transfer this NM hierarchy directly into the hierarchy of *NInChI* layers. To avoid such associations, we used tiers to describe the NM hierarchy while layers are used for the *NInChI* hierarchy as for all other notations of the InChI universe. Thus, we propose the alpha version of the *NInChI* to consist of three layers as the highest level in the hierarchy: (1) version number, (2) composition and (3) arrangement.

The first layer is a standard feature of all InChI-based notations. Since this first proposal has to be evaluated by the community and then approved by the InChI Trust, here we denote the αversion as 0.00.1A (alpha). The second layer describes sub-parts of the NM including core and shell materials but also ligands, impurities, dopants and specific linkers. For each of these components, the information consists of the chemical composition, reusing InChIs but also PInChIs or MInChIs if necessary, and additional layers to describe size, shape and additional features of categories 1 and 2 above. The third layer shows how these components are then combined to build the final NM, as shown schematically in Figure 7.

Layer 2 initially consists of the following sublayers for each component, listed in alphabetical order with “!” separating the individual components:InChI, PInChI or MInChI to represent the chemical composition (without the leading version number)morphology layer (prefix m): abbreviations are used for specific morphologies, e.g., sp: sphere, sh: shell, ro: rod, tu: tubesize layer (prefix s): specified in scientific notation in meters, e.g., 2x-9 where x can be r: radius, d: diameter, l: length, t: thickness, ranges can be given separated by “:”crystal layer (prefix k)chirality layer (prefix w for carbon nanotubes)

Layer 3 (prefix y) then specifies the overall structure, proceeding from inside and following the tiered approach specified in Section 1.2. A core-shell material is defined by 1&2, where 1 identifies the first component described in layer 2 as being the core and 2 is the second component. To indicate covalent bonding between the components, the term is enclosed in brackets. For example, (1&2&3) specifies a nano core 1 with a ligand coating 3 covalently bound using the linker 2. This definition should be flexible enough for NMs up to 2 D since they can be regarded as particles with a center even if the dimensions in one or two directions is very large. For 3 D NMs like dispersions of nanoparticles, bundles of nanowires, nanotubes and multi-nanolayers, the individual objects can be defined by *NInChI*s, but the overall structure requires another hierarchy to describe the arrangement of these objects. Nanoporous materials are somewhat special since they could be viewed as inverse NM, where the nanoform is built by the cavities. This could be described in the *NInChI* by defining the cavity as the core and then having the matrix as the first shell. However, we view this as beyond the scope of the alpha version of *NInChI* presented here.

### 3.8. NInChI Alpha—Demonstration of Worked Examples of NMs InChIs

Here we present a small number of relatively simple examples that demonstrate how the proposed *NInChI* would work in real-world scenarios. Layers are color coded in red, blue and orange for layers 1, 2 and 3, respectively for the first couple of examples, and *NInChI* specific parts are marked in bold in all examples. As for all other InChI-based notations, the goal is to provide a standard implementation to automate the generation of the *NInChI*s based on user-provided information, and extraction of features needed for specific applications of the *NInChI* (e.g., database searches, similarity assessments, and generating input for nanoinformatics studies). However, before this occurs, the proposed concepts and definitions need to be refined by feedback from the user community.

We start with two examples from Case Study 1. The first is a gold nanoparticle with an organic capping and the second is a core-shell material built from silica and gold. The generation of the different layers of these two examples are presented in detail in Figure 8.

Silica, diameter = 20 nm, with 2 nm gold coating:

*NInChI*=0.00.1A/Au**/msh/s2t10r1-9;12r2-9**!/O2Si/c1-3-2**/msp/s20d-9/k000/****y2&1 **(r1 and r2 are used to define the inner and outer radius of the gold shell) [k000 for amorphous]

CTAB-capped gold nanoparticles, diameter = 20 nm:

*NinChI*=0.00.1A/Au**/msp/s20d-9**!C19H42N.BrH/c1-5-6-7-8-9-10-11-12-13-14-15-16-17-18-19-20(2,3)4;/h5-19H2,1-4H3;1H/q+1;/p-1**/**y1&2

However, a simpler and more widely applicable possibility is defining the thickness of the shell, since this could also be applied to non-spherical cores:

*NinChI*=0.00.1A/Au/**msh/s2t-9!**/O2Si/c1-3-2/**msp/s20d-9****/**k000/y2&1

The third example is similar to the first two but not from a case study. It shows how polymer coatings can be encoded in the *NInChI*.

Polystyrene-coated silica, diameter = 20–100 nm:

*NInChI*=0.00.1A/C8H8/c1-2-8-6-4-3-5-7-8/h2-7H,1H2/z200-1-8!1S/O2Si/c1-3-2**/msp/s20:100d-9/k000/y2&1** (polystyrene is represented as the PInChI using the/z layer) [k000 for amorphous]

Case Study 2 described two examples of chiral single-wall nanotubes, the second of which is functionalized with an organic molecule. A third example from this case study was a very specific case of a functionalized fullerene C60 shown in Figure 4.

Chiral single-wall nanotube of the (3,1) type with 0.4 nm diameter:

*NInChI*=0.00.1A/C**/mtu/s4d-10/w(3,1)/y1** (even if the third layer could be omitted, we keep it to clearly denote this is a NM).

Chiral single-wall nanotubes of the (3,1) type with 0.4 nm diameter functionalized with 0.5 equivalent:

*NinChI*=0.00.1A/**C/mtu/s4d-10/w(3,1)!**C7H5F3/c8-7(9,10)6-4-2-1-3-5-6/h1-5H/**y(1&2)**

This last example only specifies that the components 1 and 2 are covalently bound but provides no information on the ratio of 1 and 2, or which atoms form the covalent bond. To include this, concepts from the MInChI (/\g layer to specify mole fractions) and the PInChI (specifying the bonding atom in the/\y layer together with the component number, here atom 1 of component 2 is bound to an unknown atom of component 1) could be adopted e.g.,:

*NinChI*=0.00.1A/**C/mtu/s4d-10/w(3,1)!**C7H5F3/c8-7(9,10)6-4-2-1-3-5-6/h1-5H/**y(1&2-1)/g{67mf-2&33mf-2}**

Note that this *NInChI* has one hydrogen too many, which needs to be removed to form the covalent bond, but we choose to keep it. In the next iteration of the *NInChI*, after wider community engagement with the InChI Trust, a better and more general description of covalent bonding can be devised. Similarly, for the CTAB-capped gold nanoparticles, a decision is still needed on how to deal with the Br- counter ion and the surface charge of the gold core.

The 5-phenylvalerate resulting in phenyl-C61-butyric acid methyl ester (PCBM) shown in Figure 4 is represented as fullerene C60 functionalized with one equivalent methyl:

*NInChI*=0.00.1A/C12H16O2/c1-14-12(13)10-6-5-9-11-7-3-2-4-8-11/h2-4,7-8H,5-6,9-10H2,1H3!C60/c1-2-5-6-3(1)8-12-10-4(1)9-11-7(2)17-21-13(5)23-24-14(6)22-18(8)28-20(12)30-26-16(10)15(9)25-29-19(11)27(17)37-41-31(21)33(23)43-44-34(24)32(22)42-38(28)48-40(30)46-36(26)35(25)45-39(29)47(37)55-49(41)51(43)57-52(44)50(42)56(48)59-54(46)53(45)58(55)60(57)59/y(2-9&1)/x{5mf-1&}

Since this example is also a well-defined molecule, it could also be represented by a standard InChI. However, the *NInChI* representation has the advantage that the common core is easier to identify. This facilitates searches for all functionalized fullerenes as described in case study 4, or could define the nanoform of functionalized fullerenes in the regulatory setting of Case Study 6.

Examples of core-shell particles from Case Study 3 have already been provided above. To demonstrate how other aspects of this case study, especially alloys and impurities, can be handled, we created a *NInChI* for the example given in the case study of doped TiO_2_ NMs.

TiO_2_ nanoparticle of 2 nm in the anatase form doped with 5% HfO_2_:

*NInChI*=0.00.1A/2O.Hf&2O.Ti/n{1&2}/g{5wf-2&}**/msp/s2d-9/k{****I 41/a m d}/y1**

(the stochastic solid mixture is represented by a MInChI and the crystalline form by its space group).

The above examples all represent single substances within experimental uncertainty e.g., in the size measurement. These substances can also be referenced to by their *NInChI* in database searches, and starting structures for nanoinformatics investigations can be created based on the information they encode. However, reiterating, the *NInChI* is not an identifier but a structural representation for a nanomaterial. Thus, additional metadata or a material-specific identifiers have to be included in searches to guarantee that the data relate to exactly for the same material, or even batch of material. This is not a limitation, rather opens up the possibility of using the *NInChI* to evaluate similarity needed for Case Studies 4 and 5, and to describe groups of substances (e.g., nanoforms) for the regulatory purposes in Case Study 6. A use case search example is if someone wants all data for gold-coated silica particles independently of their core size. The search algorithm could return all entries where the *NInChI* includes the three substrings/Au/msh, O2Si/c1-3-2/msp and/y2&1. To refine the search, especially for the regulatory use case, group representations can also be used. We present here an example taken from the “Opinion on Titanium Dioxide (nano form) as UV-Filter in sprays” released by the Scientific Committee on Consumer Safety [158]. The characteristics of the specific nanoform under which the UV-filters should be registered are specified by the Draft COMMISSION REGULATION (EU) amending Annex VI to Regulation (EC) No 1223/2009 of the European Parliament and of the Council on cosmetic products that are relevant here are: Purity ≥ 99%Rutile form, or rutile with up to 5% anatase, with crystalline structure and physical appearance as clusters of spherical, needle, or lanceolate shapesMedian particle size based on number size distribution ≥ 30 nmCoated with silica, hydrated silica, alumina, aluminium hydroxide, aluminium stearate, stearate, stearic acid, trimethoxycaprylylsilane, glycerin, dimethicone, dimethicone/methicone copolymer, simethicone;

If we neglect the coating to keep the *NInChI* simple for current discussion, and the 5% of allowed anatase that cannot be described in the *NInChI* yet, the group representation becomes:

*NInChI*=0.00.1A/2O.Ti/g{>=99wf-2}/mcl/s>=30l-9/k{**P 42/m n m**}/y1 (cl in the morphology layer/m is for cluster) or to make clear that it needs to be in the nano size range (1–100nm) defined by ECHA

*NInChI*=0.00.1A/2O.Ti/g{>=99wf-2}/**mcl/s30:100l-9/k{****P 42/m n m}/y1**

One of the batches discussed in the opinion has a median particle size of 102 nm and a purity of 99.5%, which translates to the specific *NInChI* for this substance:

*NInChI*=0.00.1A/2O.Ti/g{>=995wf-3}/**mcl/s102l-9/k{****P 42/m n m}/y1**

When comparing this substance with the second group *NInChI* above, it becomes clear that the size is not within the limits specified for the group even if it is a borderline case. This shows the advantage of encoding the upper bound for nanomaterials according to REACH directly into the *NInChI*. However, this example also makes it clear that the *NInChI* in its current form is not able to completely describe a nanoform. Characteristics like specific surface area have to be reported separately, and size ratios can only be defined as specific ranges for each dimension. If these ranges are large, the values could be chosen outside the required ratio.

These examples show that the proposed *NInChI* is indeed able to encode a variety of NMs. More complex NMs, multi-walled nanotubes, or inorganic NMs with complex ligand shells, can be represented by combining the features demonstrated by these simple examples. However, there are still limitations in what can be presented, especially in the nice-to-have category 2 (e.g., defects). Even more important, there is still no consensus on how to include the stochastic nature of NMs. Compared to small molecules, which are well defined structures, NMs often show broad distributions of the properties, including those encoded in the *NInChI*. As described in case study 1, gold nanospheres can be synthesized in very narrow and defined size ranges. This is not true for many other NMs, especially at the beginning of their design process, but the current *NInChI* does not provide a way to distinguish a well-defined Au NM with 2 nm diameter from a batch without strict size control and a broad distribution that also has an average size of 2 nm. Being able to give size ranges somewhat addresses this issue but even for such ranges, the actual size distribution might be very different and unknown. As also stressed in the case studies, the measured size depends on the method used for its determination. Regulatory guidelines therefore state that size must be measured by electron microscopy and some of the case studies requested that only sizes measured this way can be encoded in the *NInChI*. However, this would prohibit specification of this key characteristic for all but the most extensively studied NMs. In contrast, for almost all NMs an estimate of the size is available and, when combined with an uncertainty estimate, is very important information for database searches and nanoinformatics modelling. Therefore, the majority of authors also consider such less precise values essential information for a NM and supports its reporting in the *NInChI*. Further discussions with the broader community will be needed to decide if the precision of the given values for size but also concentration ratios and morphologies should be encoded in the *NInChI* or if such information needs to be collected from additional metadata as was decided for the end groups of polymers in the PInChI definition.

### 3.9. Prototype of a NInChI Generation Service

Based on the content presented in this paper and the introduction of the first *NInChI* specification, a prototype version of a *NInChI* tool has been implemented and launched (Figure 9) (http://enaloscloud.novamechanics.com/nanocommons/NInChI/). While still at an early stage and not supporting all features yet, readers and the wider scientific community can access the tool and review both the *NInChI* as well as the tool implementation. The prototype tool was built using the ZK framework employing Java in the backend. InChIs corresponding to components in the shells are retrieved using the REST API provided by https://cactus.nci.nih.gov/chemical/structure.

The tool is provided through a user-friendly interface where a non-experienced user can easily build an *NInChI* for the NM of preference. The alpha release assumes that the user is specifying the NM following the inside-to-outside structure approach and asks about information on the basic *NInChI* features like the NM core and shell(s) composition, size and the crystal space group using the Hermann Mauguin notation. The chiral indices (n, m) [159] can be provided in the case of CNTs. The shell is assumed to follow the core morphology.

Each field included in the user interface is briefly described below:

The *Composition* field refers to the NM core; different NMs can be included that can be, for example, metals (i.e., Au, Ag), metal oxides (i.e., Fe_x_O_y_) or carbon-based NMs.

For *Morphology*, the user selects one of the prespecified morphologies for the inner core (sphere, rod, cube, star, cage, prism and tube) or the unknown or complex options when applicable. When a morphology is selected, values of the size must be submitted for the specified parameters in each case.

A *Crystal structure* can also be selected, for materials where this is relevant, using the prespecified options (amorphous, TiO_2_ rutile, TiO_2_ anatase, Au, Ag, Ag_2_S, ZnO, ZrO_2_, CeO_2_, SiO_2_ Quartz and SiO_2_ or none).

*Chirality* is activated in the case of CNTs and the (n, m) values must be submitted.

When the user wishes to add outer layers on top of the inner core, a morphology of either a Shell or a Cluster can be selected to form the NM of interest. The core and the shell(s) together with their specifications are shown below the submission fields.

The *NInChI* is produced based on the submitted values and is shown on the screen below the input parameters. A full tutorial will be available and linked to the tool for more information and guidance. Figure 9 shows the generation of the *NInChI* for an Au spherical NM with a core diameter of 25 nm encapsulated by an Ag shell of external diameter 30 nm resulting in:

*NInChI*=1A/Ag/msh/s30t-9/k [Fm-3m]!Ag/**msh/s30t-9/k[Fm-3m]/y2&1**

## 4. Conclusions

While there are valid concerns that the InChI concept struggles with substances that have only partially defined structures, such as complex chiral chemicals, inorganics, and coordination complexes, it remains the leading approach for structural representation of chemicals. Indeed, extensions for polymers have been proceeding for 10 years, and there is encouraging progress towards the establishment of an InChI for NMs, or *NInChI*. For example, fullerene already has a well-defined InChI, suggesting that this can work for NMs, although it is debatable whether fullerene is really a large molecule or a small NM.

The 8-month process of brainstorming, iterating and reflecting, and development of three NMs-specific and three user-perspective case studies, identified the main features of NMs that should be captured by an *NInChI*, and proposed the layers into which this information should be organized. This culminated in a *NInChI alpha version*, for discussion and further refinement with the NM and nanoinformatics communities and the InChI Trust. The proposed *NInChI alpha* adopts existing conventions, integrates aspects from MInChI and RInChI, uses an additive approach to incorporate existing InChIs for small molecule surface functionalisations, and aims to complement the existing suite of extensions to InChI. A key decision made by the group was to focus on the intrinsic properties of NMs only, rather than including dynamic properties and behaviors that depend on the surrounding conditions, extrinsic NMs properties. Although these are also important, we propose to consider them in future versions of the *NInChI*.

The first layer is a standard feature of all InChI-based notations indicating the version number. The second layer describes subparts of the NM including core and shell materials but also ligands, impurities, dopants and specific linkers. For each of these components, the information given in this layer consists of the chemical composition, reusing existing InChIs, PInChIs and MInChIs where available, and additional layers to describe NM size, shape, crystal structure or chirality (where relevant) and surface features, defects, etc. The third layer then shows how these components are combined to build the final NM, starting from the inside and working outwards towards the reactive surface. One issue that remains to be addressed is how to capture covalent bonds more effectively, e.g., for CNTs.

NMs features or properties that were considered beyond the current scope include nanocomposite and nanostructured materials, inverse NMs (nano holes in a bulk material), and nanoporous materials. Dynamic properties such as dissolution, agglomeration, and protein corona formation were also excluded for now, although it seems likely that these could be added using a modified reactions extension of InChI, in which reactions are considered to be transformations.

The *NInChI alpha* offers huge potential to drive FAIR nanosafety data, facilitate nanoinformatics workflows, grouping and regulation of NMs. All of the NMs features identified as distinguishing “nanoforms” under the REACH regulatory approach are captured in this proposed *NInChI alpha* and as such we are confident of strong support from across the research, industrial and regulatory communities to implement this under the auspices of the InChI Trust, and we look forward to stimulating discussions on the next steps and further iterations of the *NInChI*.

## Figures and Tables

**Figure 1 nanomaterials-10-02493-f001:**
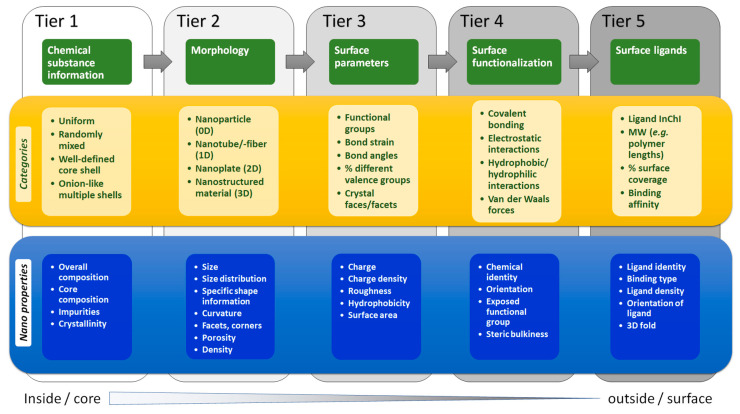
Proposed hierarchy (from inside out) of five tiers needed to describe the chemical and structural complexity of engineered NMs, with different tier categories (yellow panels) and NM-related properties (blue panels). While most published literature does not provide detailed characterization of all tiers (especially Tiers 3 and 5) it is becoming clear that such information is necessary to connect NMs properties to their effects [41]. Computational approaches may be very useful here by estimating surface properties and their relationships to toxicity, and for differentiating different nanoforms * and the boundaries between sets of nanoforms (* see the regulatory case study for further details on nanoforms [42]).

**Figure 2 nanomaterials-10-02493-f002:**
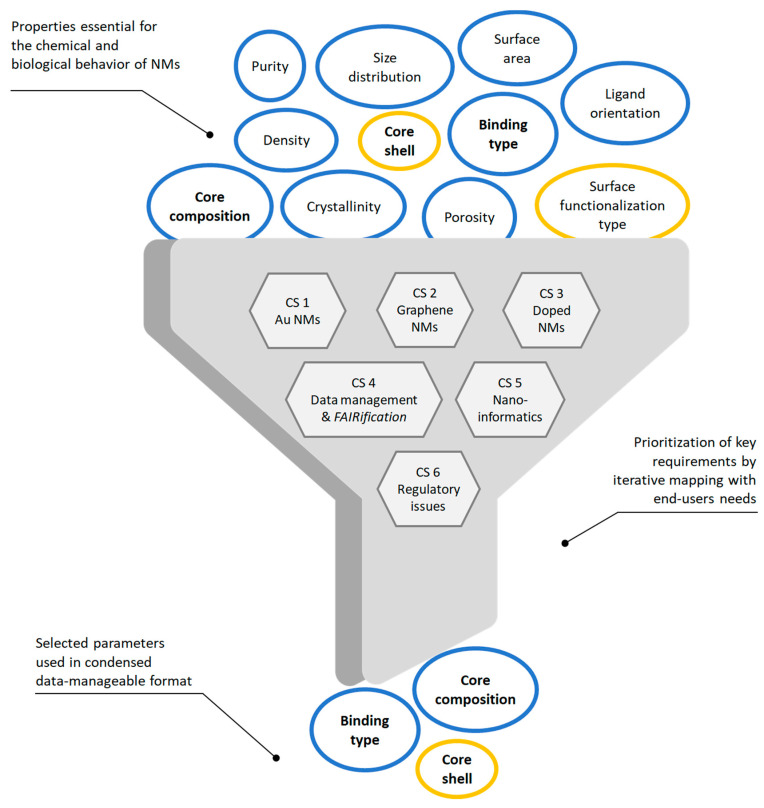
Schematic of the approach to determine the essential components of the *NInChI* from the six case studies that encode the NM-specific aspects identified by experimentalists (CS 1–3) by iteratively mapping them to the needs of the end-users (CS 4–6). This prioritization process recommended the layers essential for a structural notation for NMs that allow discrimination between unique nanoforms of the same material (tier categories in orange, NM-related parameters to be determined/as produced in blue, with detailed descriptions as per Figure 1).

**Figure 3 nanomaterials-10-02493-f003:**
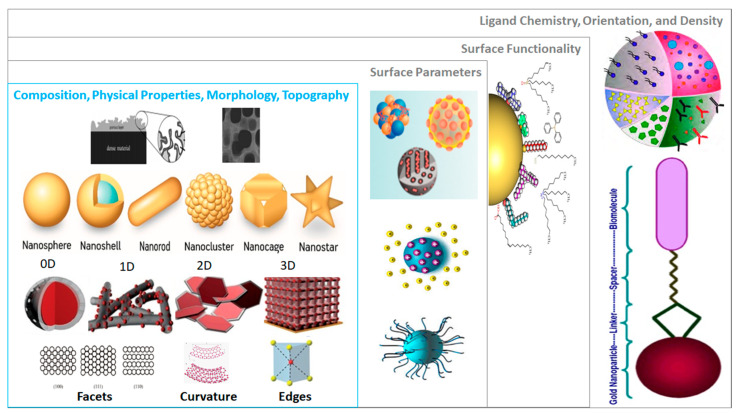
Overview of characteristic properties of Au-based NMs highlighted in case study 1—dimension/shape of core, size and its distribution, surface functionalization, and types of bonding. Chemical composition, physical properties, morphology, and nanotopography of the core (light blue text) correspond to Tiers 1 and 2, surface parameters, functionality, ligand chemistry, orientation, and density (in gray) correspond to Tiers 3–5 of Figure 1.

**Figure 4 nanomaterials-10-02493-f004:**
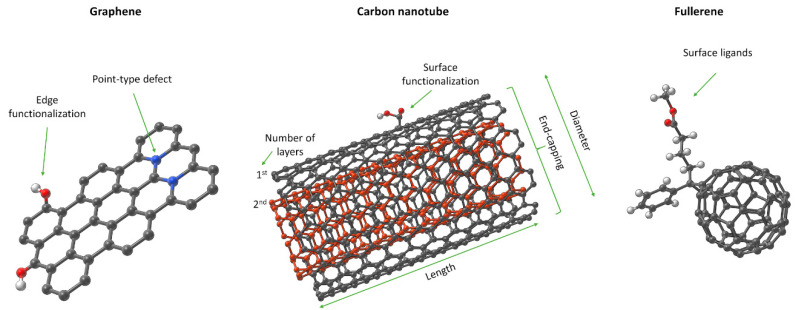
Characteristic properties of graphene family NMs highlighted in case study 2.

**Figure 5 nanomaterials-10-02493-f005:**
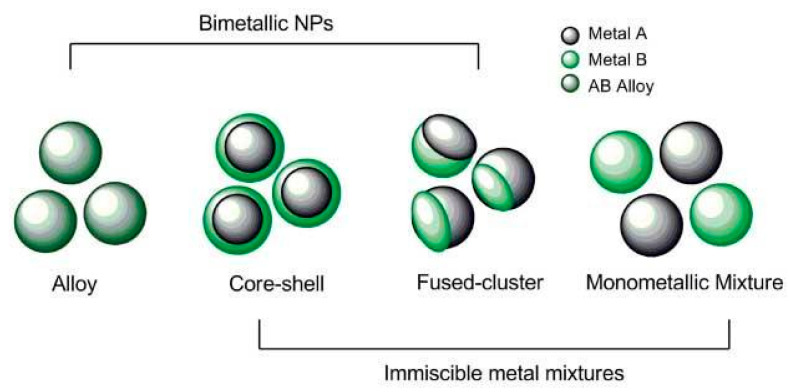
Possible outcomes from bimetallic NM synthesis—similar structures will also result from multi-metallic NMs. The same chemical formula would apply to each of the first 3 materials viz. MetalA_x_MetalB_1−x_O_y_. Structural representation must capture morphological differences in tier 1 of Figure 1 [130].

**Figure 6 nanomaterials-10-02493-f006:**
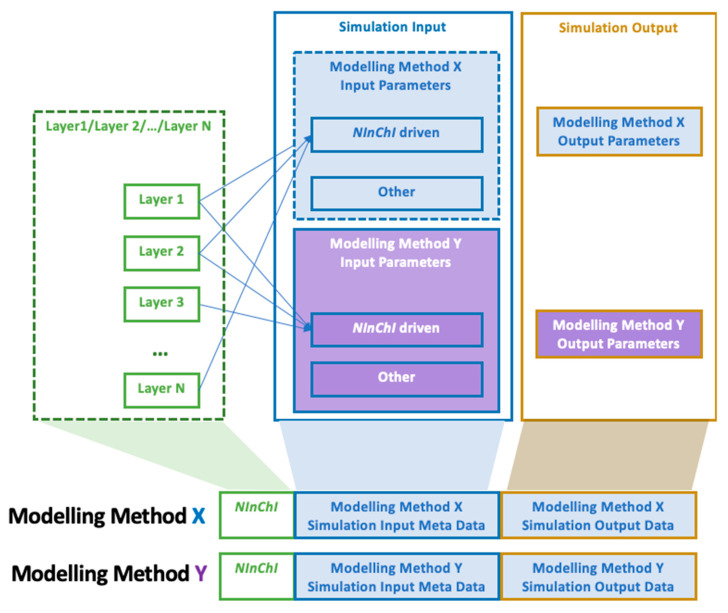
The same *NInChI* is associated with different metadata for simulation, specific to modelling methods X and Y, leading to computational outputs X and Y such as structural or energetic descriptors as indicated in Table 2.

**Figure 7 nanomaterials-10-02493-f007:**
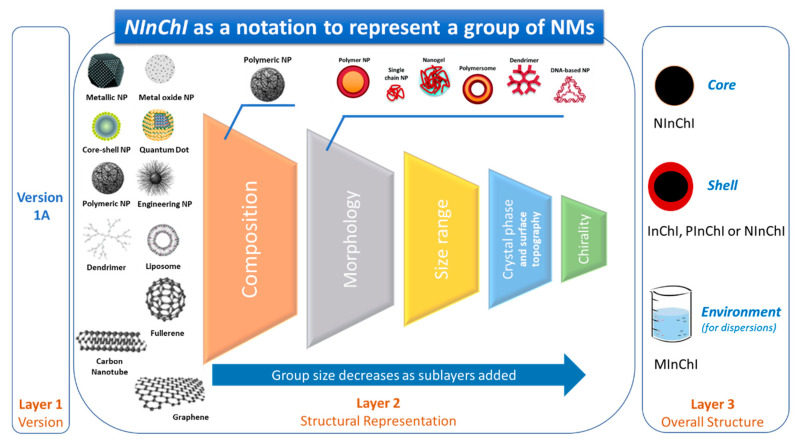
Illustration of the *NInChI* as a notation to represent a particular group of NMs, with the group size decreasing as additional sub-layers and NM components (core, shell(s), surroundings in the case of NMs provided as dispersions) are added. Note that the *NInChI* incorporates existing InChIs for small molecules, PInChIs for polymeric coatings and MInChI for the wt% of particles in dispersion).

**Figure 8 nanomaterials-10-02493-f008:**
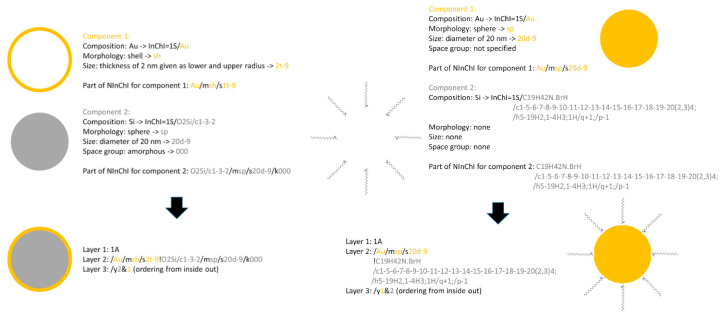
Creation process of the NInChI starting with the generation of the parts for layer 2 for the individual components, which are then combined with later 1 and 3 to the final NInChI. **left**: silica nanoparticles with gold coating, **right**: CTAB-capped gold nanoparticles.

**Figure 9 nanomaterials-10-02493-f009:**
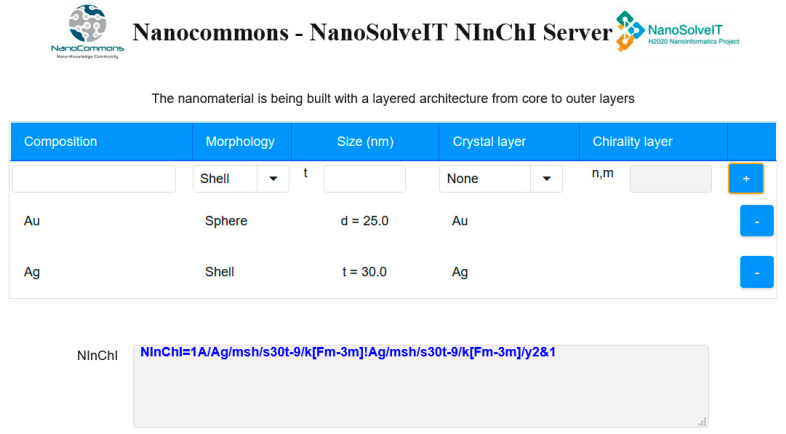
The user interface of the prototype of the *NInChI* application, where the user can build simple NM structures containing the NM’s core and potential shells and clusters. Information to be provided includes the chemical composition, crystal structure, morphology and size. The chiral indices (n, m) for CNTs can be provided also.

**Table 1 nanomaterials-10-02493-t001:** Example: Input required for atomistic simulations or evaluation of descriptors for gold NMs, and their corresponding *NInChI* elements.

Required Simulation Input	Information Encoded in the *NInChI*
Core material chemistry	Au (Gold)
Size	20d-9 (20 nm)
Shape	sp (sphere)

**Table 2 nanomaterials-10-02493-t002:** Example for Atomistic Simulations for nanodescriptors calculation.

*NInChI*	Simulations Input	Simulations Output
Core/size/shape/polyform	The structure (i.e., coordinates of all atoms) of a NM and input parameters (Buckingham and Coulomb force field parameterization)	Structural and energetic descriptors of the NM

**Table 3 nanomaterials-10-02493-t003:** Categories of NM properties captured in *NInChI* and those considered out of scope.

Category 1: Must Have	Category 2a: Nice to Have	Category 2b: Extrinsic Properties	Category 3: Out of Scope
Chemical composition Size/size distribution Shape Crystal structure Chirality Ligand and ligand binding	Structural defects Density Surface composition	Surface charge Corona Agglomeration state Dispersion	Optical properties Magnetic properties Chemical state/oxidation state

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
