# Peer review of "Can an InChI for Nano Address the Need for a Simplified Representation of Complex Nanomaterials across Experimental and Nanoinformatics Studies?"

_nanomaterials, 2020, doi:10.3390/nano10122493_

Round 1

Reviewer 1 Report

The paper presents a proposal for the first specification (an alpha version) for NInChI linear notation - an InChI extension for nanomaterials (NMs) taking into account complexity of NMs and their chemical and structural properties. The NInChI design is developed by means of methodological approach deployed and verified on the base of six types of case studies covering prospective NInChI stakeholder groups.
The volume of the manuscript is too big which makes the article hard to read and embrace, however the complexity and importance of the topic justifies the large number of pages. Enormous of work is done (long process of brainstorming, iterating, reflecting, and development) by various domain experts in different scientific fields collaborating together for efficient handling of the NMs information.
The paper makes thorough analysis of the information needed for building relationships between the NMs’ characteristics and their properties. The authors have identified the major challenges for NInChI implementation, the required design for NInChI layers and the needed syntax and semantic capabilities of the future NInChI realization. The major achievement of the paper is the proposed NInChI design which goes beyond the classical graph based representation, typical for linear notations, and capturing the complexity of NMs by means of 5 tires in the NInChI. The proposed tier organization for substance info, morphology, surface parameters, functionalization and ligands is adequate, efficient and logically robust, combining the expertize from multiple domains. Another important consideration for the future NInChI implementation is the efficiency of NMs data management and FAIRness as well as the usefulness for different categories of stakeholders.
To me, this article is a kind of a road map which could be very helpful for NM scientists and nanoinformatics specialists to navigate within the “jungle” of huge amount of information associated with the NMs. The paper also could be very useful in the context of good practices for NMs’ data managing and especially for data generators and nanoinformatics community - one of the primer users of NMs data.
This work is a very import milestone towards the realization of efficient linear notation for representation for nanomaterials (NInChI). The authors have succeeded to find a good balance between the need to capture as many as possible NMs characteristics and the practical software implementation for everyday use with minimum number of descriptors.
I hope that Nanoinformatics and Chemoinformatics communities will soon be able to implement the proposed NMs data design in a NInChI linear notation supposedly using most (if not all) of the features and design suggested in this paper.
The paper could be accepted for publication after addressing the major and minor issues listed below:

Major issues:
1. General suggestion: shorten the paper by moving parts of text into the supplementary materials, if possible. The authors have done great study and summaries which should be available to the scientific community. One reasonable approach is to move significant part of section 3 (case studies details) into the supplementary. Another way to optimize the text is to remove repeating thoughts and ideas (there is a lot of stuff mentioned several times in different use cases) as well as well-known and established principles in nanoinformatics are repeated or explained in too much detail.

2. Section 3.4.2 / parts I and II: in some sentences NInChI is addressed as an “identifier”, for example: rows 1015-1019: “The physicochemical properties included in the layers of the new NInChI identifier will make data more findable and enhance scientific interoperability by allowing exclusion of NMs that meet some but not all of the criteria contained in the identifier (e.g., only TiO2 NMs that have at least 80% anatase and a specific surface functionalization)”
I think it is not correct to name NInChI “identifier” in this context. Identifiers can be used only for the purpose of exact searching of chemical objects or for NMs registering (i.e. NMs sameness context is OK for term “identifier”) but not for NM filtration and obtaining NM hit-lists matching given criteria e.g. “at least 80% anatase” or “exclusion” of NMs. In this statement, NInChI should not be called “identifier” but it must be stressed out that it is a linear notation describing/encoding rich chemical information about the NM. Also, the NInChI info must be parsed and processed in order to be used for more complicated types of searching different than “identity search”.
The same holds for the concluding statement of part II (rows 1082-1086) – term “identifier is ok for the first part (“sameness”) but not for the second part “similarity”:
“➤ If a NInChI is to support establishing “sameness” of NMs (across batches, following storage or ageing, etc.) this identifier should additionally methods for determining the relevant properties to ensure direct comparability / interoperability. We therefore argue for a NInChI that encodes sufficient information to quickly gauge “similarity” that is “good enough” in many applications requiring integration of measurements from multiple batches and samples of NMs.”
Another instance of incorrect usage of term “identifier” is in rows 1118-1119:
“…it will become possible to find, mine, and process data according to any specific NM search query (as long as it is encoded in the identifier).”
Also, I think in figure 7, “NInChI as a group identifier” is the same conceptual misuse of the term “identifier”. “NInChI represents, encodes or describes particular group” is better statement.
Generally, I would suggest within entire paper, NInChI to be addressed/called “linear notation” which in various cases could be used (or function) as an “identifier”. Please check all places where NInChI is called “identifier” improperly (not all instances are incorrect, some are ok); there is a kind of mismatching of term “identifier” occasionally.

3. I am confused with the summarizing statement of section 3.6.2/part I (rows 1330-1333):
“By acting as compound IDs (rather than substance IDs) NInChI will support the differentiation of individual nanoforms (assessment entities) and Tiers 1-3 in Figure 1 map to the composition, size/shape and surface coating which are the distinguishing features of nanoforms”.
Generally, the NMs are substances, not chemical compounds as it is the concept of REACH regulation. What do authors mean by “acting as compound IDs (rather than substance IDs)”? I think this statement contradicts to the REACH approach and definition of substances versus compounds (the latter may be components of the substances/NMs)

Minor issues:

1. Author Tomasz Puzyn is duplicated (listed twice) in the author list. The same duplication is done in the supplementary materials as well.
2. row 54: canonical/unique SMILES can be considered as chemical identifier but not the SMILES notation in general. Also using SMILES as a chemical identifier across different data bases and platforms is a bit tricky since different canonicalization algorithms might be used.
3. row 81: Complex sentence: “During this time, a transition from pure in vivo toxicology studies to more mechanistic in vitro studies involving advanced organotypic cell culture models mimicking the relevant human body barriers, i.e. inhalation and ingestion, occurred.”. I would suggest to split it and make it more clear.
4. row 187: Small typo: “An RInChI” should be “A RInChI”
5. row 378: Small typo in “For example, molecular dynamics simulations characterized…”. Word “characterized” should be “characterize”
6. row 433: “…have distinctly different intrinsic different chemical properties.”: word “different” is repeated
7. row 489: in “…adapted from the for reaction, polymer, mixture”, word “for” to be removed
8: row 504: Please specify abbreviations “CTAB” and “CIT”
9. row 668: reference [83] is placed within word “microscopy”
10. row 669: in sentence “However, in the literature provides multiple ways to report size and shape …”  word “in” is not needed or another correction is required.
11. row 897: “… applied to complex NMs, there are specific additional characteristics that need to be considered”. Word applied should be corrected to “apply” or “are applied” or something else…?
12. row 1167: the meaning of abbreviation “QC” is not given.
13. I would suggest in Table 1: replace the “X”-es with real examples (values / MODA excerpts for particular NM). This way, table 1 would be more useful.
14. The same suggestion for Table 2. I think it would be more efficient for the readers to see real simulations input and output. Otherwise this table just could be replaced with a single sentence.
15. rows 1242-1244. “The bioinformatics space has clearly shown the advantage of such an approach, with the great advances occurring in biopharmaceuticals, health research, and basic science being driven by advanced computational tools that were only made possible the bioinformatics space is searchable and interoperable”. Please correct this complex sentence. It is not clear or correct(some small word might be missing?).
16. row 1288: In sentence “Similarly, a representation of substance and reference compound for NMs must also incorporated into IUCLID, …” Probably word “be” is missing in front of “incorporated”
17. row 1355: “…the range within which a specific property of the NMs can vary for their and toxicity to..”, Please check the wording, “for their” is not ok?
18. row 1398: “see Table 5” should be corrected to “see Table 3”

Reviewer 2 Report

The authors propose the creation of a unique identifier for nanomaterials based on the successful InChI identifiers for small molecules. Their proposed NinChI-alpha represents a first step in this direction, which the authors see as basis for discussion that would end in the definition of a standardized unique identifier. The nature and diversity of nanomaterials adds additional layers of complexity compared to standard InChIs. As such, it cannot be expected that a canonicalized NInChI would be able to perfectly describe all kinds of nanomaterials.
The authors do a very job in presenting six case studies focusing on different aspects. The first three focus different types of materials while the latter three focus on the usefulness and application of NiChIs from a database, nanoinformatic, and regulatory point of view. This makes their rationale easy to follow. The discussion describes a their NInChI-alpha and gives some examples. While it does not amount to a formal specification, it is explains the proposed structure of the identifier well.
I think, the representation could be improved, by giving a layer by layer description of one or two of the NInChI examples given (lines 1505-1520), maybe in the form of a figure. (Similar ones can be fount in the InChI specifications.)
However, I consider this a very minor issue and recommend "publication as is".

Author Response

The authors thank the reviewer for their constructive feedback on our manuscript, which we are confident has helped to improve the clarity and readability of the manuscript.  We are confident that this paper will make an important contribution to the field and as such ensuring readability and precision are essential.

Comments

The authors do a very job in presenting six case studies focusing on different aspects. The first three focus different types of materials while the latter three focus on the usefulness and application of NiChIs from a database, nanoinformatic, and regulatory point of view. This makes their rationale easy to follow. The discussion describes a their NInChI-alpha and gives some examples. While it does not amount to a formal specification, it is explains the proposed structure of the identifier well.

Author response:  We thank the reviewer for the positive feedback on the structure and approach and are really pleased that the logic and approach was clear to you as a reader.

I think, the representation could be improved, by giving a layer by layer description of one or two of the NInChI examples given (lines 1505-1520), maybe in the form of a figure. (Similar ones can be fount in the InChI specifications.)

Author response: We thank the reviewer for this very constructive comment, and have revised this section to provide a better mapping to the case studies in terms of the exemplar NInChI presented. We agree that this makes the logic and flow of the paper clearer.

However, I consider this a very minor issue and recommend "publication as is".

Authors reply: We sincerely thank the reviewer for his/her appreciation of our work! As we have performed a number of revisions in the manuscript, we have also implemented the requested figures to make the NInChi representation clearer for readers.

Reviewer 3 Report

Authors present a very interesting work in the area of knowledge representation of chemoinformatics and nanoinformatics.

The paper presents an extension of InChI for representing nanoparticles and nanomaterials, and discusses the benefits of using such extension. The extension is composed of three layers for notation, and five tiers for describing different characteristics of nanomaterials.

Article is in general well written and well presented.

I have some comments and suggestions regarding format to be adressed:

  • Introduction contains some methodological parts, I recommend to move all the text describing the five tiers of NInChi into the Materials and Methods Section (part of 1.2). Also the description of the approach based in case studies (1.3)
  • Figure 2 does not provides any information, in my opinion. Its textual explanation provides much more information than the Figure itself.
  • Reference [83] is wrongly located in text, appearing in the middle of a word.
  • I strongly recommend to format use cases description from Material and Methods Section as subsections, similarly to subsections used to discuss each use case in Result secion, for easing the reading.
  • I feel that Table 3 could be improved by adding information about the NInChi tiers together with the different categories status. Since Figure 1 introduces and compiles all possible categories and properties that could be gathered for a nanoparticle, Figure 3 should show current status of the work already done over Figure 1.
  • The three layers definition appears in Discussion Section, it should be moved to Material and Methods Section.
  • Table 1 does not provide any additional information, consider removing.
  • In the second paragraph of Conclusions Section, there is a missing point. "... should be organized This culminated ..."

Regarding the contents of the research, I think the authors performed a huge and nice work for defining the five tiers for nanoparticles formalization. But I have some concerns about some parts of the paper:

  • In final paragraph of sub-section 1.2 authors say: "Here we propose the first specification of NInChi" and they also say that its alpha version is under review by community. This first release should be provided, as a link to facilitate its access to final readers, or as supplemental material. Definition of categories and properties are not enough, final result should be accessible, so 'coded' categories and values can be accessed.
  • Use cases results are too theoretical, specially the use case 4 (FAIRification), 5 (Nanoinformatics) and 6 (regulatory), since using any standarized nomenclature or representation should benefit in the same way for those cases. Thus, concrete examples should be provided using NInChi to test the validity of hypothesis which are presented in the text. 
  • The seven coding examples provided in sub-section 4.2 are useful, but they are not used for demostration of any of the use cases.

And finally as a curiosity, authors propose an homogeneous way to represent nanoparticles and materials based in InChi, which has became a de facto standard in the field. NInChi representation is based in InChi and its extensions for notation, to be human and computer readable. But since this notation is just a "final representation", explaining of the internal representation (model implemented in computer) of different tiers, and different substances or chemical componens, should be included, since it would be very interesting for readers.

Round 2

Reviewer 3 Report

Authors have improved the report addressing some of comments in previous version.

Imho methodology and results section contents are still mixed. While authors claim that the structure of NInChi is the result of their research, that is clear, the article itself is presenting NInChi, thus, I think that all description of internal structure of NInChi should be explained in methods Section.